# Excessive copper impairs intrahepatocyte trafficking and secretion of selenoprotein P

Maria Schwarz [1,2,16], Caroline E. Meyer [1,2,16], Alina Löser [1,2], Kristina Lossow [1,2], Julian Hackler[2,3], Christiane Ott [2,4], Susanne Jäger[2,5], Isabelle Mohr[6], Ella A. Eklund[7,8], Angana A. H. Patel[7,8], Nadia Gul[7,8], Samantha Alvarez[7,8], Ilayda Altinonder [7,8], Clotilde Wiel[7,8], Maria Maares [2,9], Hajo Haase [2,9], Anetta Härtlova[8,10,11], Tilman Grune[2,4], Matthias B. Schulze [2,5,12], Tanja Schwerdtle[2,12,13], Uta Merle[6], Hans Zischka [14,15], Volkan I. Sayin[7,8], Lutz Schomburg [2,3] & Anna P. Kipp [1,2] ✉

Selenium homeostasis depends on hepatic biosynthesis of selenoprotein P (SELENOP) and SELENOP-mediated transport from the liver to e.g. the brain. In addition, the liver maintains copper homeostasis. Selenium and copper metabolism are inversely regulated, as increasing copper and decreasing selenium levels are observed in blood during aging and inflammation. Here we show that copper treatment increased intracellular selenium and SELENOP in hepatocytes and decreased extracellular SELENOP levels. Hepatic accumulation of copper is a characteristic of Wilson's disease. Accordingly, SELENOP levels were low in serum of Wilson's disease patients and Wilson's rats. Mechanistically, drugs targeting protein transport in the Golgi complex mimicked some of the effects observed, indicating a disrupting effect of excessive copper on intracellular SELENOP transport resulting in its accumulation in the late Golgi. Our data suggest that hepatic copper levels determine SELENOP release from the liver and may affect selenium transport to peripheral organs such as the brain.

Selenium is an essential nutrient and trace element that is of vital importance for human health. Twenty-five human genes encode for selenoproteins, all of which contain selenocysteine (Sec) residues at their active sites which are catalyzing redox reactions[1]. Among these, selenoprotein P (SELENOP) is the most important one for distributing selenium from the liver to organs such as the brain and testes to maintain their selenium supply and local selenoprotein expression[2–4]. Approximately 60% of the total selenium concentration in serum is bound to SELENOP. Upon loss of SELENOP, mice develop severe seizures and ataxia due to selenium deficiency in the brain[5,6]. Receptors of the LRP (low-density lipoprotein receptor-related protein) family including LRP8 (APOER2) are mediating SELENOP endocytosis[7]. Unlike other selenoproteins that carry single Sec residues, SELENOP contains up to ten, one at its N-terminal domain with redox activity and nine at its

C-terminal domain which is cleaved off to release selenium within target cells[8]. While uptake of SELENOP by target tissues has been extensively studied, less attention has been paid to the release of SELENOP from hepatocytes. Ingested selenium is first transported to the liver and either used for local selenoprotein expression or for synthesis of SELENOP for further distribution within the body. *SELENOP* transcription is mediated by many factors including inflammatory mediators[9,10]. SELENOP is a glycoprotein which is further processed by both N- and O-glycosylation[11]. Newly synthesized glycoproteins destined for export such as SELENOP are translocated into the endoplasmic reticulum (ER) for early processing and quality control before further glycosylation and excretion via the Golgi apparatus takes place[12].

Besides selenium, the liver is also the main organ for maintaining homeostasis of additional trace elements such as copper. The ATP-

driven transporter ATP7B (ATPase copper transporting β) is responsible for balancing hepatic and systemic copper levels. Under basal conditions, copper is shuttled by antioxidant 1 copper chaperone (ATOX1) to the trans-Golgi and bound to ATP7B to metallate newly synthesized ceruloplasmin (CP), the major copper containing protein in serum. Holo-CP, which binds up to six copper atoms, operates as a carrier that distributes copper to peripheral organs[13]. Excess copper results in ATP7B translocation from the Golgi to endo-lysosomal compartments to store copper or to release it into the bile[14,15]. We have previously shown that copper inhibits hepatic selenoenzyme activity of the glutathione peroxidase (GPX) and thioredoxin reductase (TXNRD) families both in vitro and in vivo[16]. Thus, an adequate selenium supply could be limited on the functional level of selenoenzyme activity by copper. Circulating selenium and copper concentrations are negatively correlated, e.g. during aging copper levels are increasing while selenium levels are decreasing[17]. During inflammation, the same picture emerges as CP is an acute phase protein increasing in the circulation while SELENOP is a negative acute phase reactant declining in the circulation[10,18]. SELENOP is a candidate target protein for interference of copper with selenium homeostasis as it represents the main route of selenium excretion from the liver. Based on this, we hypothesized a direct effect of copper on SELENOP excretion resulting in the inverse regulation of the trace elements in hepatocytes and blood. Accordingly, we aimed to study whether copper excess affects intracellular selenium levels, and SELENOP distribution and excretion of cultured hepatocytes and whether the findings are compatible with data from genetic models of ATP7B mutations in rats and Wilson's disease patients.

Here we show that copper treatment increases intracellular selenium and decreases extracellular SELENOP levels using the liver-derived cancer cell line HepG2. The same effect is observed in cultured primary murine hepatocytes and supported by an untargeted secretome approach in which SELENOP is particularly downregulated by copper. Accumulation of copper in liver is a characteristic of Wilson's disease and LPP rats, which serve as a disease model. Accordingly, SELENOP levels are relatively low in serum of patients with Wilson's disease and LPP rats, especially in those with low CP concentrations. The CP promoter contains the SNP rs11708215, which is associated with increased serum copper and CP concentrations. An analysis of a large cohort study (EPIC-Potsdam) shows a congruent positive correlation of SNP rs11708215 (coded per G allele) with serum SELENOP levels. Overall, the results indicate that genetic variations modulating hepatic copper concentrations cause changes in the amount of circulating SELENOP which most probably affects the selenium distribution to peripheral organs such as the central nervous system and endocrine glands.

## Results

### Copper blocks SELENOP excretion from HepG2 cells

To study the cross-talk of the trace elements selenium and copper, HepG2 cells were cultured for up to 72 h with $CuSO_4$ or $Cu(His)_3$ in combination with selenite or selenomethionine (SeMet). None of the chosen copper concentrations in combination with or without 50 nM selenite showed cytotoxic effects (Supplementary Fig. 1a)[16]. The copper treatment with $CuSO_4$ (Fig. 1a) or $Cu(His)_3$ (Supplementary Fig. 1b) resulted in a more than 20-fold increase of cellular copper levels which was independent of the selenium supply of the cells. Both selenite and SeMet treatment upregulated the cellular selenium concentration by a factor of 4 (Fig. 1b, S1c). In previous studies, we identified an overall reduction of enzymatic activity of selenoproteins by copper[16]. Unexpectedly and in contrast to the reduced selenoprotein activity, selenium levels were increased in copper-treated cells by a factor of 1.5 (Fig. 1b, S1c). This effect was independent of the copper and selenium source. For further experiments, mainly selenite and $CuSO_4$ were used as selenium and copper sources, respectively.

HepG2 cells excrete SELENOP into the medium. A 24 h incubation with increasing copper concentrations up to 500 μM, resulted in a

concentration-dependent decrease of SELENOP excretion independent of the selenium status (Fig. 1c). Also upon copper treatment for 72 h, extracellular SELENOP concentrations were reduced (Fig. 1d, upper Western Blot; Supplementary Fig. 1d) as indicated by the copper-induced fold changes of 0.3-0.7 (Fig. 1e, white bars). In parallel, SELENOP accumulated within copper-treated cells (Fig. 1d, lower Western Blot) which was quantified as a fold change of 1.3 and 1.7 for selenite- and SeMet-treated cells, respectively (Fig. 1e). This effect was much more pronounced for fully glycosylated SELENOP with a molecular weight of ~65 kDa represented by the upper band in Western Blot analyses (Fig. 1d, blue box, upper band) which was quantified as fold change of 2 and 2.7 for selenite- and SeMet-treated cells, respectively (Fig. 1e, S1e). In contrast to the intracellular SELENOP accumulation, qPCR analyses revealed a mild downregulation of *SELENOP* mRNA while *LRP8* mRNA was slightly upregulated by copper (Fig. 1f). However, protein concentrations of the SELENOP transporter LRP8 in the membrane/organelle fraction were not modulated by copper treatment (Supplementary Fig. 1f).

To exclude that the observed effects are restricted to HepG2 cells, a 24 h ex vivo copper treatment of primary murine hepatocytes was performed resulting in intracellular copper accumulation and reduced extracellular SELENOP concentrations compared to cells without copper treatment (Fig. 1g, h). When using colon-derived HT-29 cells which do not excrete measurable amounts of SELENOP, copper treatment and concomitantly higher intracellular copper levels did not increase but even decrease intracellular selenium concentrations (Supplementary Fig. 1g, h) indicating that the copper effect on selenium accumulation depends on SELENOP and its accumulation. To further test this hypothesis, an siRNA-mediated knockdown of *SELENOP* was generated in HepG2 cells. Both extracellular and intracellular SELENOP levels were substantially reduced in *SELENOP* knockdown cells (Fig. 1i, j). In addition, neither a copper-induced effect on extra- or intracellular SELENOP (Fig. 1i, j) nor on cellular selenium concentrations (Fig. 1k) was detectable upon knocking down *SELENOP*. Downregulation of *SELENOP* did not change the cytotoxicity of copper levels up to 1 mM in comparison to cells expressing *SELENOP* (Fig. 1l). Overall, the data indicate that copper increased the intracellular selenium content by blocking SELENOP excretion which appears to be specific for liver cells.

To characterize the time-course of the copper-related inhibition of SELENOP excretion, the incubation time with selenium and copper was reduced to 6 h. Already after 6 h of treatment, measurable amounts of SELENOP were excreted into the medium which were reduced by copper (Fig. 2a, b). In parallel, selenium started to accumulate inside the cells (Fig. 2c). To reverse the copper-induced effects on SELENOP excretion, we treated the cells 24 h prior to harvest with the two copper-specific chelators BCS and TTM[16]. This treatment partially rescued the inhibition of SELENOP excretion by copper (Fig. 2d). However, chelator treatment only without additional copper was not able to increase SELENOP excretion. To further test for the specificity of the inhibition of SELENOP excretion by copper, additional cations, namely zinc and iron, were used. In contrast to copper, zinc treatment reduced both intracellular and extracellular SELENOP concentrations indicating that zinc might inhibit SELENOP expression (Fig. 2e, S2a). This effect was confirmed on mRNA level, where downregulation of *SELENOP* and upregulation of *LRP8* was even more pronounced after zinc than after copper treatment (Supplementary Fig. 2d). Fold changes were still relatively small (0.6 for *SELENOP* and 1.4 for *LRP8*) in comparison to the strong upregulation of metallothionein *MT2A* mRNA expression in response to zinc but also to copper with fold changes of 90 and 20, respectively (Supplementary Fig. 2e). Iron increased intracellular SELENOP protein concentrations, but did not affect mRNA levels, intracellular selenium or extracellular SELENOP levels indicating that iron might affect cellular turnover of SELENOP (Fig. 2e, Supplementary Fig. 2b). As all three cations are able

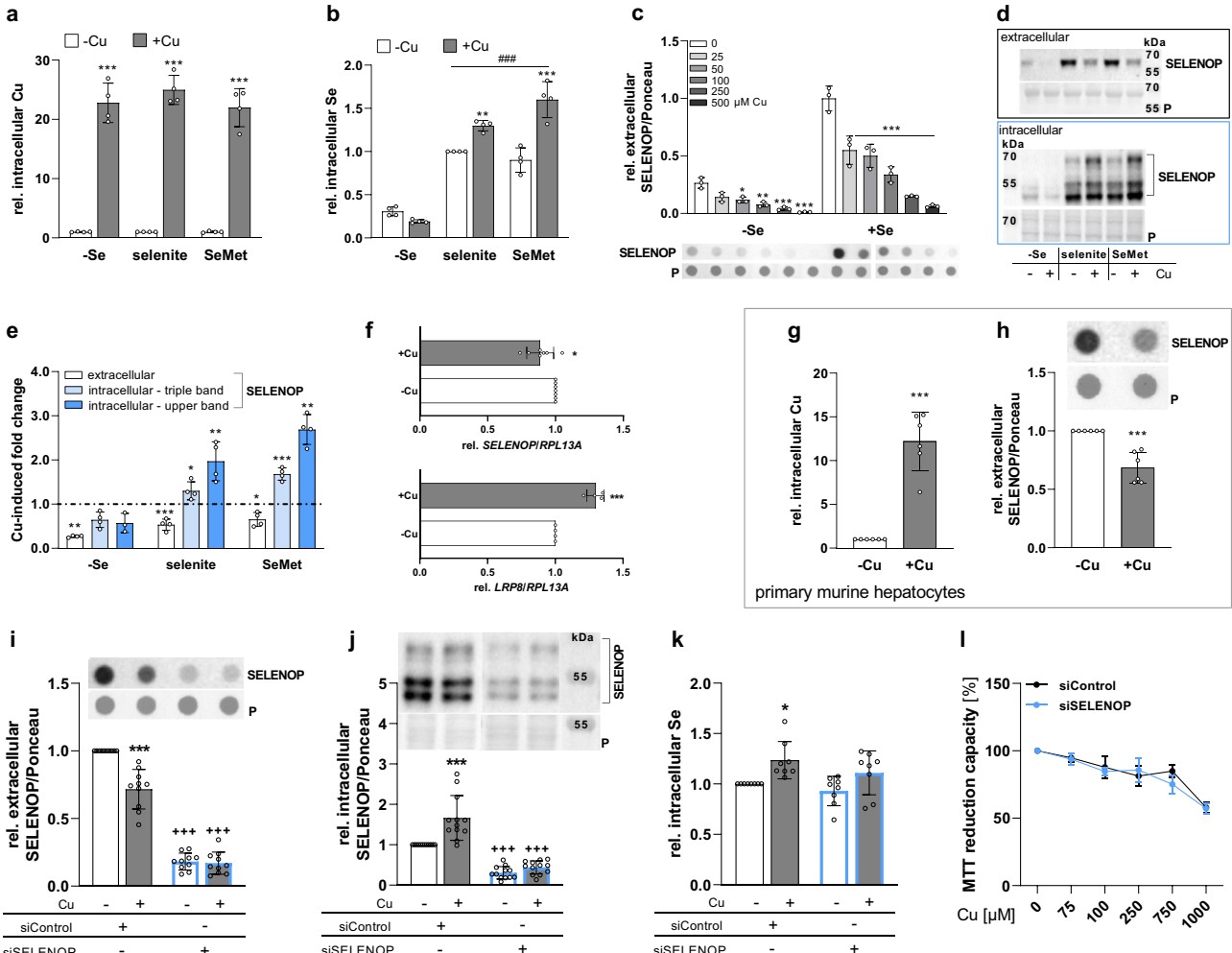

**Fig. 1 | Copper interferes with hepatic selenium homeostasis.** Intracellular Cu and Se concentrations were determined using TXRF with 1 mg/L Yttrium as standard for 1000 s and normalized to protein content of HepG2 cells (**a**, **b**, **g**, **k**). Extra- and intracellular SELENOP was determined using Western or dot blot and normalized to Ponceau (P) staining (**d**, **e**, **h**–**j**). **a**, **b** Intracellular Cu and Se content of cells treated with 0 or 100 µM CuSO₄ without (-Se) or with 50 nM selenite or 200 nM SeMet (*n* = 4). **c** Extracellular SELENOP was determined in medium of HepG2 cells treated for 72 h with or without selenite and increasing Cu concentrations 24 h prior to harvest (*n* = 3). **d** Representative Western Blots of extra- and intracellular SELE-NOP and Ponceau staining of cells treated as described for **a**–**b**. **e** Quantification of extra- and intracellular SELENOP bands from Western Blots (shown in **d**) presented as Cu-induced fold changes (*n* = 4). Respective cells without Cu treatment were set as 1 (dotted line). **f** mRNA expression of SELENOP (*n* = 7) and LRP8 (*n* = 4) was determined by qPCR after 24 h incubation with 0 or 100 µM CuSO₄ and normalized

to RPL13A. **g**, **h** Intracellular Cu concentration and extracellular SELENOP of primary, murine hepatocytes treated for 24 h after isolation with 50 nM selenite in combination with 0 or 10 µM CuSO₄ (*n* = 6). **i**–**k** A siRNA-mediated knockdown of SELENOP was generated and extra- (*n* = 10) and intracellular SELENOP (*n* = 12) and intracellular Se concentrations (*n* = 8) were analyzed after 72 h treatment with selenite and/or Cu. **l** MTT reduction capacity of SELENOP knockdown and control cells treated for 48 h with selenite including a 24 h incubation with increasing Cu concentrations. Cytotoxicity was related to cells without Cu treatment (*n* = 3). Data are depicted as mean ± SD. Biological replicates are indicated by individual dots. Statistical analyses were based on two-way ANOVA with Bonferroni's post-test (**a**–**c**, **i**–**l**) or on two-tailed *t* test compared to cells without Cu treatment (**e**–**h**). \**p* < 0.05; \*\**p* < 0.01; \*\*\**p* < 0.001 vs. -Cu, ###*p* < 0.001 vs. -Se; +++*p* < 0.001 vs. siControl. Source data are provided as a source Data file.

to modulate the cellular redox state towards oxidation, we further treated cells with H₂O₂ to mimic this effect. As observed for zinc, a decrease of both extra- and intracellular SELENOP was detected in H₂O₂ treated cells (Fig. 2e, Supplementary Fig. 2c). Thus, the identified inhibitory effect on SELENOP excretion was specifically observed for copper only and does not appear to be driven by a general modulation of the cellular redox status.

**Hepatic SELENOP excretion is suppressed by copper in vivo**
Based on the cell culture data, we aimed to test for in vivo relevance of our results. Accordingly, we were wondering whether a hepatic copper accumulation which was studied in cell culture would affect SELENOP excretion. Hepatic copper accumulation takes place in Wilson's disease caused by mutations of ATP7B. The LPP rat is a Wilson animal

model with genetic ablation of ATP7B and growing rats develop different stages of Wilson's disease (classified as affected, disease onset, and diseased) over time, as described earlier[19,20]. Already at the early, affected stage, copper strongly accumulated in the liver of homozygous rats which was maintained in all disease stages (Fig. 3a). The two early stages were characterized by lower plasma copper concentrations (Fig. 3b) and very low CP oxidase activity in plasma (Fig. 3c) in comparison to heterozygous control rats. In the diseased stage, circulating copper concentrations increased again back to levels of control rats due to damage of liver tissue (Fig. 3b) while CP concentrations stayed low (Fig. 3c). Selenium and SELENOP levels were downregulated during the course of the disease (Fig. 3d, e). Plasma levels of CP and SELENOP were positively correlated (Pearson's *r* = 0.522; *p*-value = 0.011). The treatment of diseased LPP rats with the

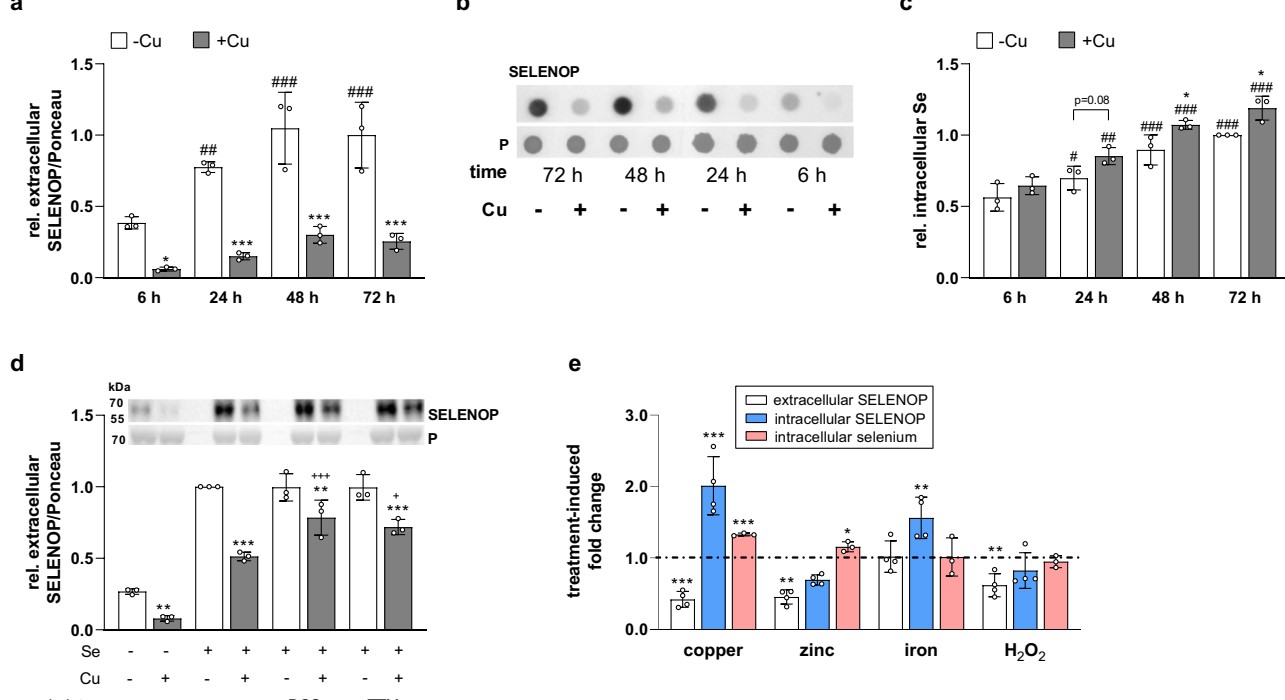

**Fig. 2 | Copper is a fast inhibitor of SELENOP excretion. a–c** Extracellular SELE-NOP and intracellular Se concentrations of HepG2 cells which were co-incubated with Cu and Se for indicated time points, analyzed using dot blot and normalized to Ponceau (P) staining or by TXRF ($n = 3$). **d** Extracellular SELENOP of HepG2 cells treated for 72 h with 0 or 100 μM CuSO₄ in combination with or without 50 nM selenite. In addition, 24 h prior to harvest the Cu chelators BCS or TTM were added. Samples were analyzed by Western Blot, normalized to Ponceau staining, and selenite-treated cells were set as 1 ($n = 3$). **e** Extra- and intracellular SELENOP ($n = 4$) and selenium concentration ($n = 3$) of HepG2 cells treated with 50 nM selenite in combination with either 100 μM Cu, Zn or Fe for 72 h or with 600 μM H₂O₂ for 6 h (in selenite supplied cells) presented as fold change to selenite only treated cells. Respective cells without further treatment were set as 1 (indicated by the dotted line). Data are depicted as mean ± SD. Biological replicates are indicated by individual dots. Statistical analyses were based on two-way ANOVA with Bonferroni's post-test (**a, c, d**) or on two-tailed $t$ test compared to untreated cells (**e**). *$p < 0.05$; **$p < 0.01$; ***$p < 0.001$ vs. -Cu (**a, c, d**) or other treatments such as -Zn, -Fe or -H₂O₂ (**e**), #$p < 0.05$; ##$p < 0.01$; ###$p < 0.001$ vs. 6 h; ++$p < 0.01$; +++$p < 0.001$ vs. -chelator. Source data are provided as a source Data file.

copper chelator methanobactin reduced circulating copper concentrations (Fig. 3f) as expected[20] and accordingly resulted in higher circulating selenium and SELENOP concentrations (Fig. 3g, h). Based on these in vivo results combined with the in vitro studies, we can clearly conclude that copper interferes with selenium homeostasis and hepatic SELENOP excretion.

## Relationship of copper and selenium serum biomarkers in humans

We next analyzed the selenium status in serum samples of Wilson's disease patients at the time point of diagnosis before the onset of therapeutic interventions. Overall, the clinical parameters of the patients were very heterogeneous, e.g., a very broad range of serum copper concentrations was observed. Based on their serum CP, the patients were divided into two groups with normal (>90 μg/L) or low (<90 μg/L) serum CP concentrations (Fig. 4a) which was confirmed by measuring the functional holo-CP reflected by CP oxidase activity (Fig. 4b). Low CP is one of the parameters used for the diagnosis of Wilson's disease[21] which most probably indicates a higher hepatic copper accumulation as shown for the rat Wilson model (Fig. 3). In addition, a third group ('diseased') was established for those patients with high clinical serum parameters such as C-reactive protein (CRP) and hepatic transaminases indicating already a more severe disease state resulting in copper leakage from destroyed hepatocytes (for patient's characteristics see table S1). To further characterize the patients' copper status, total and free copper were analyzed. In line with the CP concentrations, total copper levels were reduced in the 'low CP' group in comparison to the 'normal CP' group (Fig. 4c). Free copper levels confirmed the division of patients into the 'diseased'

group with higher concentrations in comparison to the two other groups (Fig. 4d). The SELENOP concentration was significantly decreased in the serum of patients with low serum copper and CP concentrations in comparison to the 'normal CP' group but was heterogeneous within the 'diseased' group (Fig. 4e). The overall serum selenium concentration showed the same picture as for SELENOP but was also reduced in the 'diseased' group (Fig. 4f).

CP is the main transport protein for copper from the liver to peripheral organs, meaning that low circulating CP levels would indicate that copper is retained in the liver. The SNP rs11708215 is located in the promoter region of the *CP* gene and has been described to enhance hepatic CP expression and CP secretion[22] which would as a consequence result in more efficient copper export from the liver and a concomitant decrease in hepatic copper levels. The analysis of EPIC-Potsdam samples revealed that the SNP rs11708215 (coded per G allele) was positively correlated with serum levels of copper[23] but also with SELENOP (Fig. 4g). This indicates that more SELENOP can be excreted from the liver when hepatic copper levels are low.

## Copper modulates the secretome of HepG2 cells and strongly interferes with SELENOP release

To explore the mechanisms behind copper-induced effects on blocking SELENOP excretion we used the HepG2 cell model. The overall effects of copper treatment on the secretome of HepG2 cells were analyzed by quantitative LC-MS/MS. The analysis led to the identification of 1,179 proteins in the medium of HepG2 cells. Gene ontology (GO) enrichment analyses revealed that downregulated proteins were involved in glycosaminoglycan, heparin, and sulfur compound binding and possess oxidoreductase activity (Supplementary Fig. 3a).

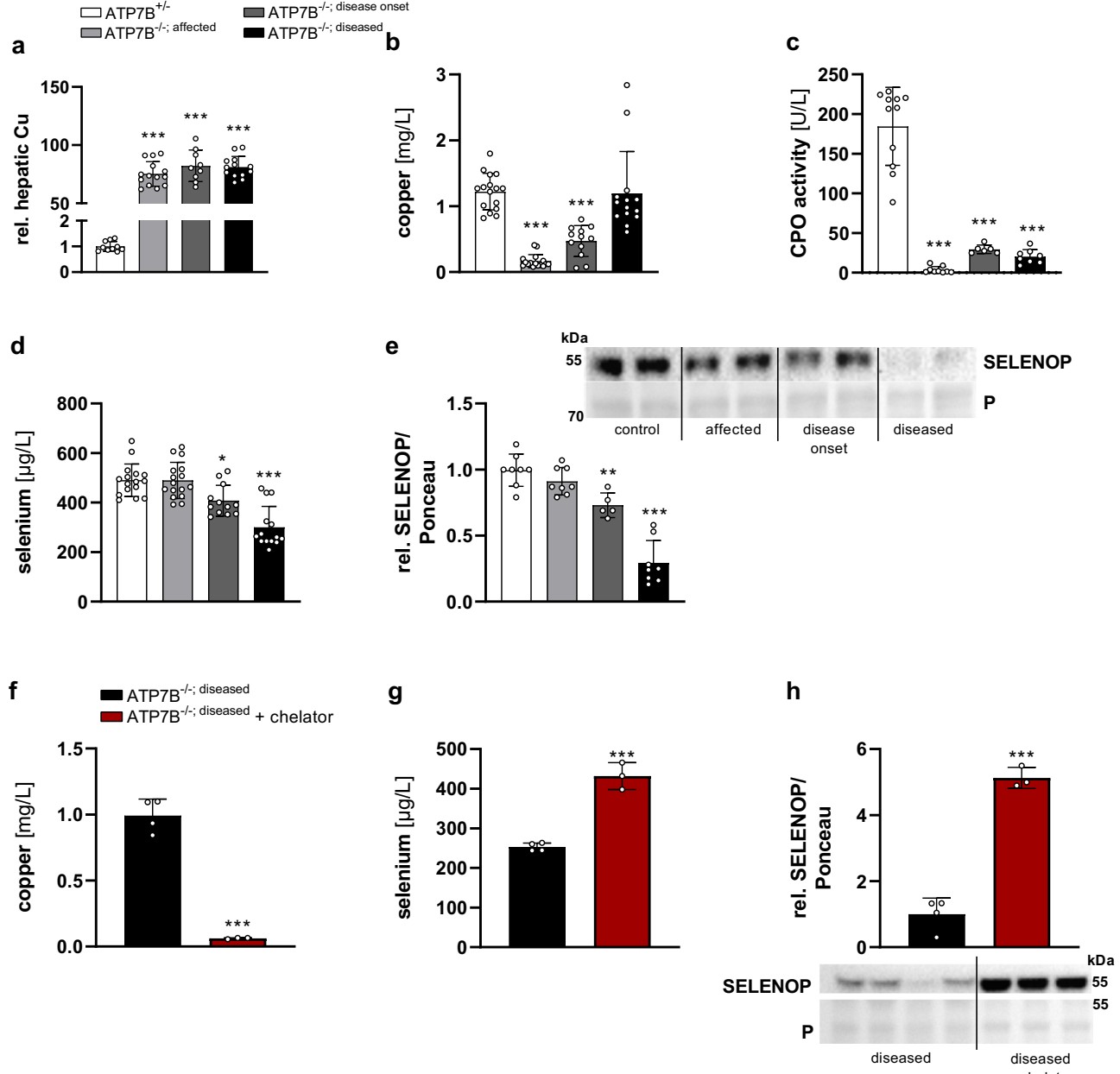

**Fig. 3 | Copper effects on selenium homeostasis in LPP rats. a** Hepatic and (**b**, **f**) plasma Cu and (**d**, **g**) Se concentrations of female and male control rats (ATP7B[+/−]) and ATP7B[−/−] at different stages of Wilson's disease (affected, disease onset, diseased) ($n = 6–16$) or (**f**–**h**) 8 days after application of the Cu chelator methanobactin ($n = 3–4$) were measured using TXRF with 1 mg/L yttrium as standard element for 300 s and normalized to protein content or with 1 mg/L gallium as standard element for 1000 s, respectively. **c** The CP oxidase (CPO) activity in plasma of rats was measured photometrically. **e**, **h** Plasma SELENOP was analyzed by Western Blot and normalized to Ponceau (P) staining ($n = 5–8$). Data are depicted as mean ± SD. Biological replicates are indicated by individual dots. Statistical analyses were based on one-way ANOVA with Bonferroni's post-test *$p < 0.05$; **$p < 0.01$; ***$p < 0.001$ vs. ATP7B[+/−] control rats (**a**–**e**) or by two-tailed $t$ test compared to untreated rats (**f**–**h**). Source data are provided as a source Data file.

Upregulated proteins also had oxidoreductase and hydrolytic activity, were involved in carbohydrate binding and belonged to the peroxiredoxins. The GO enrichment for cellular compartments confirmed the reliability of the results mainly representing proteins located to exosomes or vesicles (Supplementary Fig. 3b) including SELENOP. Copper treatment significantly upregulated 195 and downregulated 75 proteins which together accounted for 23% of the detected proteins in the secretome. SELENOP was one of the most significantly and consistently downregulated proteins in response to copper treatment (Fig. 5a). Overall, the secretome analysis showed that there was no general inhibition of secretory processes by copper. Only 6.4% of the whole secretome were downregulated in the extracellular space in response

to copper. One of the upregulated proteins was CP, the main extracellular copper binding protein[13]. The release of α1-antitrypsin (AAT, SerpinA1) a highly abundant liver-derived serum protein[24] was unaffected by copper (Supplementary Fig. 3c, Supplementary Data 1) in agreement with the majority of secreted proteins in this analysis.

Next to SELENOP, proteins of the apolipoprotein family (APOB, E, and H) were consistently downregulated as well. For APOE, we could confirm an early extracellular decrease after 6 h of copper treatment (Fig. 5b) and a concomitant intracellular APOE accumulation (Fig. 5c) showing the same pattern for APOE as observed for SELENOP. APOE has been described to bind to SELENOP and to block its excretion[25]. Thus, the copper-induced intracellular APOE accumulation may

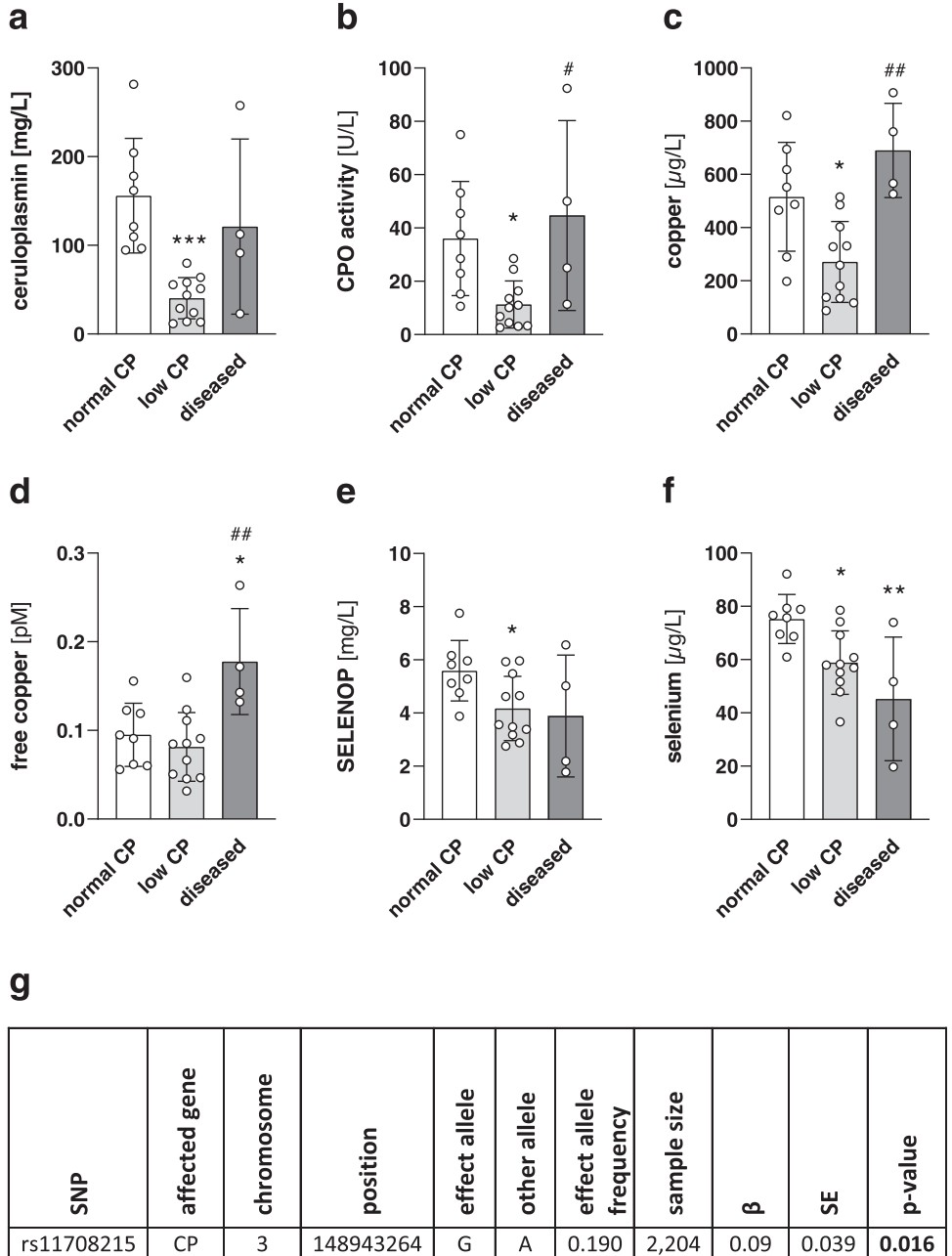

**Fig. 4 | SELENOP concentrations are positively correlated with serum copper.**
**a** Serum ceruloplasmin (CP) concentrations of Wilson's patients at time point of diagnosis measured by ELISA. Serum CP oxidase (CPO) activity (**b**), total copper (**c**), free copper (**d**), SELENOP (**e**), and selenium (**f**) concentrations of Wilson's patients measured using photometric activity, TXRF with 1 mg/L gallium as standard element for 1000 s or a fluorometric method (normal CP $n = 8$; low CP $n = 11$; diseased $n = 4$). Data are depicted as mean ± SD. Biological replicates are indicated by individual dots. Statistical analyses were based on one-way ANOVA with Bonferroni's post-test. *$p < 0.05$; **$p < 0.01$; ***$p < 0.001$ vs. 'normal CP', #$p < 0.05$; ##$p < 0.01$ vs. 'low CP'. **g** Results from EPIC-Potsdam samples for SNP rs11708215 (coded per G allele) and its correlation with serum SELENOP from linear regression using an additive genetic model adjusted for age at recruitment and sex. Serum SELENOP levels were natural log-transformed and standardized. Source data are provided as a source Data file.

contribute to the reduced SELENOP excretion. Based on this, we generated an *APOE* knockdown in HepG2 cells reducing intracellular APOE levels to 60% of controls (Fig. 5d). SELENOP excretion was substantially increased upon *APOE* knockdown under basal conditions but was still reduced by copper treatment (Fig. 5e).

**SELENOP accumulates in the late Golgi in response to copper treatment**

As secretory proteins are processed in the Golgi, we used the inhibitors brefeldin A and monensin to interfere with intracellular protein shuttling. Brefeldin A blocks the transport of secretory proteins from the

ER to the Golgi complex, and thus completely abolishes Golgi-resident glycosylation processes[26]. Accordingly, no distinct glycosylation of intracellular SELENOP was detectable in brefeldin A-treated cells (Fig. 6a). There was no copper-dependent effect on intracellular SELENOP or selenium levels after brefeldin A treatment (Supplementary Fig. 4a, b). Next, monensin was used to inhibit protein transport from the medial to the trans Golgi complex[27]. Monensin-treated cells showed strong intracellular SELENOP accumulation which, however, was restricted to the two lower bands of SELENOP (Fig. 6b). The fully glycosylated form of SELENOP with a size of 65 kDa which was most sensitive to copper treatment (Fig. 1d, e) was not detectable after

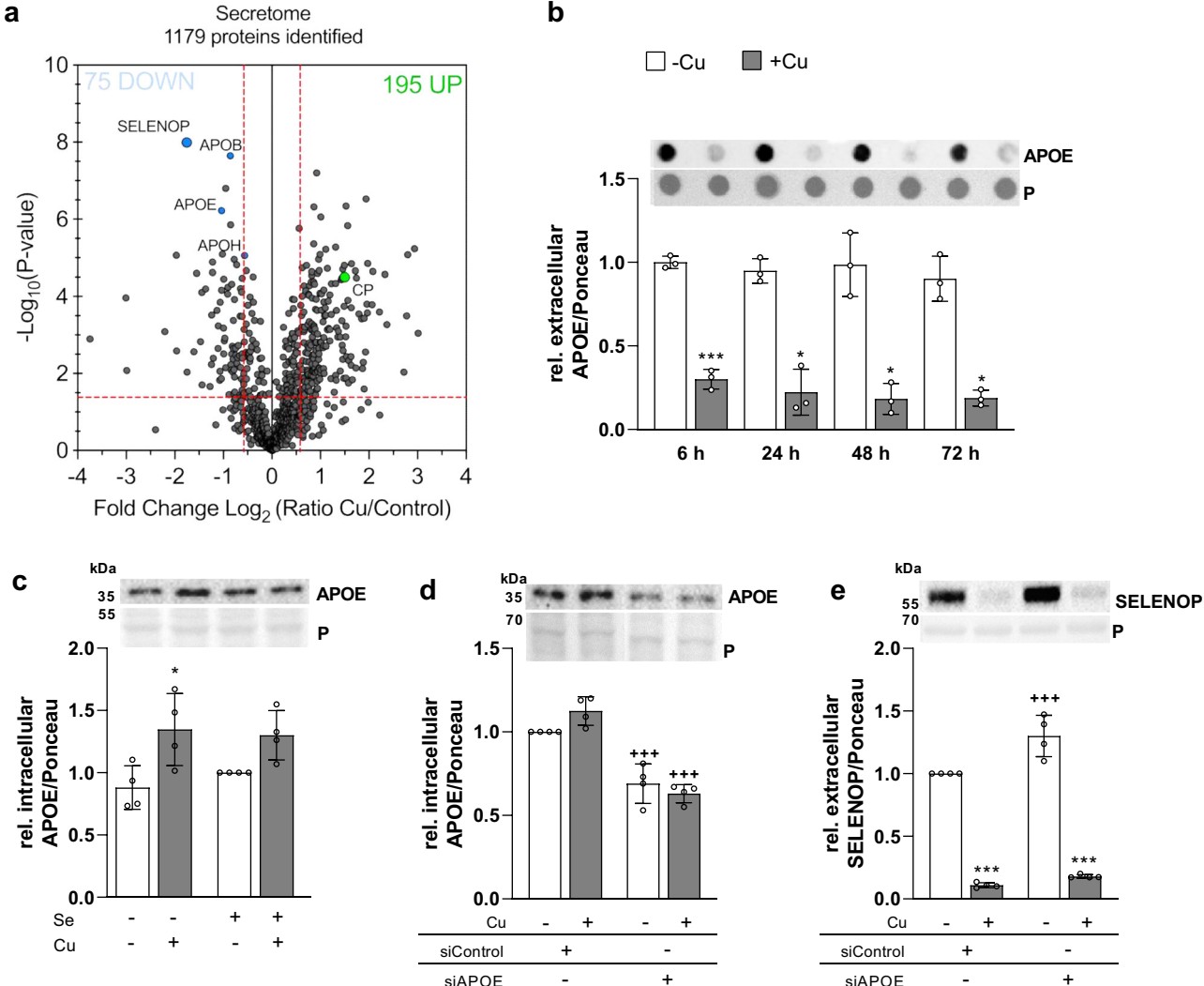

**Fig. 5 | Copper modulates the HepG2 cell secretome. a** Volcano plot of proteins identified by secretome analysis in the medium of HepG2 cells cultured for 48 h in the presence of 50 nM selenite and with 0 or 100 μM CuSO₄. Two-sided t-test-based statistics were applied on normalized and logarithmized protein ratios to extract the significantly regulated proteins. **b** Extracellular APOE of HepG2 cells co-cultured with Cu and Se for indicated time points. APOE was analyzed using dot blot and normalized to Ponceau (P) staining (*n* = 3). **c** Intracellular APOE in HepG2 cells treated with 0 or 100 μM CuSO₄ without (-Se) or with 50 nM selenite for 72 h measured by Western Blot normalized to Ponceau staining (*n* = 4). An siRNA-mediated knockdown of *APOE* was generated and intracellular APOE (**d**) and extracellular SELENOP (**e**) were analyzed after 72 h treatment with selenite and/or Cu by Western Blot normalized to Ponceau staining (*n* = 4). Data are depicted as mean ± SD. Biological replicates are indicated by individual dots. Statistical analyses were based on two-way ANOVA with Bonferroni's post-test (**b**–**e**). Adjustments for multiple comparisons were made for data provided in (**a**). *$p < 0.05$; **$p < 0.01$; ***$p < 0.001$ vs. -Cu; ++$p < 0.01$; +++$p < 0.001$ vs. siControl. Source data are provided as a source Data file.

monensin treatment indicating that this glycosylation step takes place in the trans or late Golgi. In line with this, copper treatment did not increase intracellular SELENOP but even decreased SELENOP and selenium levels in monensin-treated cells (Supplementary Fig. 4c, d). To further specify the compartment where the copper-sensitive gly-cosylation of SELENOP takes place, deglycosylation experiments using PNGase F and EndoH were performed[28]. PNGase F digestion cleaves off all N-glycans and accordingly results in the deglycosylation of all three SELENOP forms (Fig. 6c). A comparable lack of glycosylation was observed when treating cells with tunicamycin which inhibits N-linked glycosylation[29] and accordingly also substantially reduced SELENOP glycosylation (Supplementary Fig. 4e). In tunicamycin-treated cells, copper reduced intracellular SELENOP levels (Supplementary Fig. 4e) as observed in monensin-treated cells. In contrast, EndoH more speci-fically cleaves off early Golgi N-linked sugar chains, while late Golgi N-linked sugars are EndoH resistant. The copper-sensitive glycosyla-tion of SELENOP turned out to be EndoH resistant (Fig. 6d) and

accordingly appears to be established in the late Golgi (Fig. 6e). Cel-lular fractionation experiments revealed that the Cu-induced SELENOP accumulation was undetectable in the cytosol but was enriched in the membrane/organelle fraction which also contained GOLGIN97, an established marker for the trans Golgi network (TGN) (Fig. 6f). This accumulation was detectable already after 1 h of copper treatment and increased over time (Supplementary Fig. 4f). In parallel with SELENOP, glycosylated APOE increased in the membrane/organelle fraction (Fig. 6f, Supplementary Fig. 4g). Immune fluorescence experiments also identified a copper-induced increase of SELENOP in the peri-nuclear region showing a co-localization with GOLGIN97 (Fig. 6g, h). These results indicate that copper interferes with the late Golgi cargo process resulting in accumulation of SELENOP.

## Discussion
We provide herein data from cell culture, rodent models, and Wilson's disease patients indicating strong evidence for a connection between

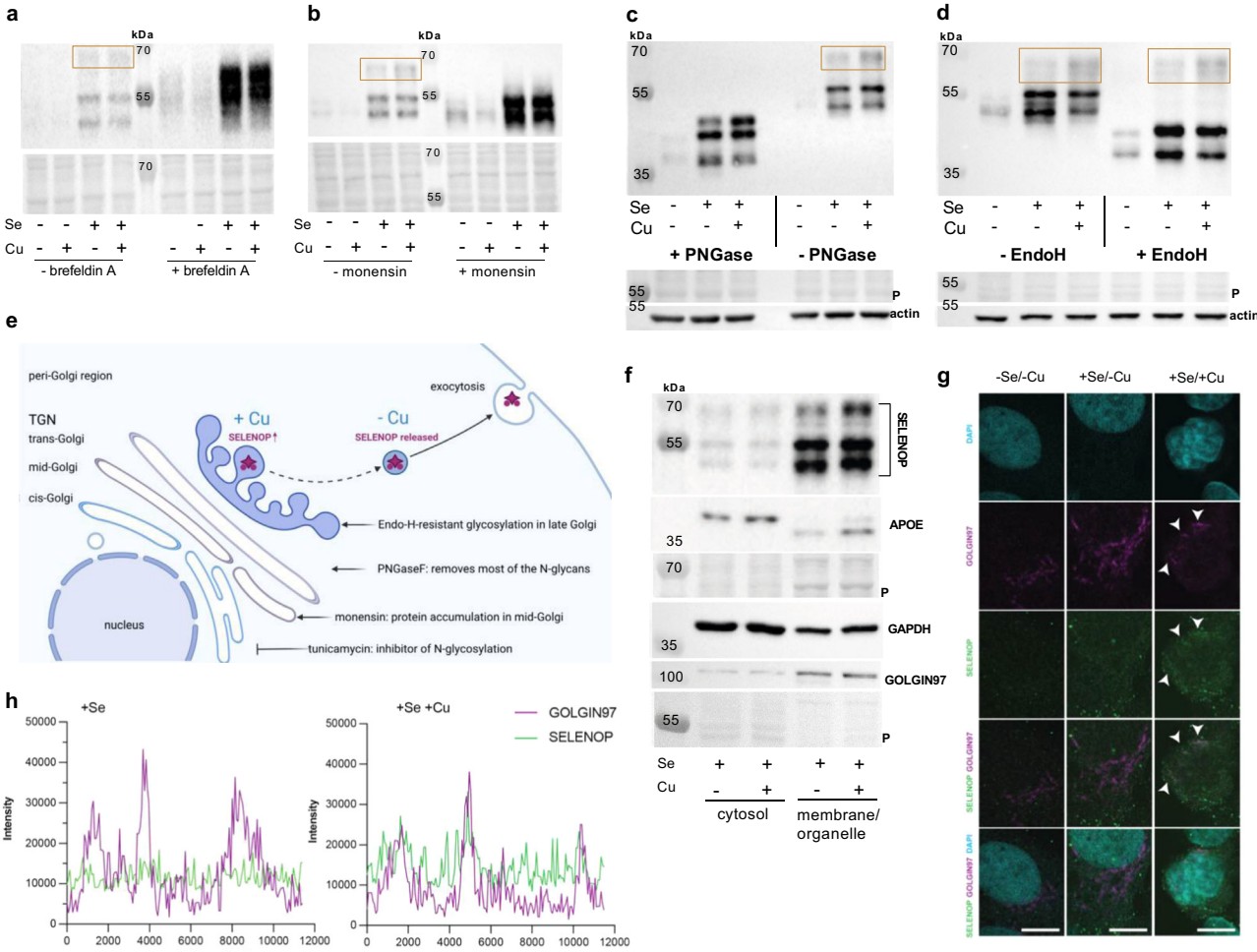

**Fig. 6 | Copper results in SELENOP accumulation in the Golgi.** Intracellular SELENOP (**a**, **b**) of HepG2 cells treated with 0 or 100 μM CuSO₄ without (-Se) or with 50 nM selenite (+Se) for 72 h. In addition, 24 h prior to harvest brefeldin A (**a**) or monensin (**b**) were added. Samples were analyzed by Western Blot normalized to Ponceau staining. Orange boxes indicate fully glycosylated SELENOP. **c**, **d** Cell lysates of HepG2 cells treated with or without 50 nM selenite in combination with 0 or 100 μM CuSO₄ for 72 h were incubated with PNGase F (**c**) or Endonuclease H (EndoH) (**d**) and intracellular SELENOP was determined using Western Blot. **e** Effects are depicted in a scheme as described in the text (TGN−trans golgi network). The figure was created with BioRender: https://app.biorender.com/. **f** SELENOP, APOE, GAPDH, and GOLGIN97 were analyzed in cytosolic or membrane/ organelle fractions of cells treated with 50 nM selenite in combination with or without 100 μM CuSO₄. Representative images are shown for all Western Blots. All experiments were repeated independently for at least three times with similar results. **g** Immunofluorescence for SELENOP (green), GOLGIN97 (violet) and DAPI (blue) was performed in HepG2 cells treated with or without 50 nM selenite in combination with 0 or 100 μM CuSO₄ for 24 h. Representative images are shown. The scale bar indicates 10 μm. Experiments were repeated independently for at least three times with similar results. **h** The co-localization of SELENOP and GOL-GIN97 was observed under +Se/+Cu conditions. Source data are provided as a source Data file.

copper and selenium homeostasis. We have consistently shown that hepatic copper accumulation reduces SELENOP excretion from hepatocytes which represents about 60% of circulating SELENOP[2]. Copper treatment resulted in lower SELENOP concentrations in cell culture media as well as in the circulation of whole organisms (Fig. 7). The untargeted secretome approach performed in the culture medium of HepG2 cells identified SELENOP as one of the most strongly down-regulated proteins in response to copper (Fig. 5a). The comparison of the secretome of HepG2 cells with that of primary human hepatocytes indicated a high overlap[24]. Among the secreted proteins several previously described hepatokines including SELENOP have been detected[24]. Also in murine hepatocytes, SELENOP release was reduced by copper (Fig. 1h). SELENOP has an essential role as selenium distributing protein to organs such as brain and testes[2–4]. Lowering circulating SELENOP concentrations has major consequences for the selenium supply of these peripheral organs which is most important in populations with a suboptimal selenium supply, which also prevails in the European population[30]. As Wilson's disease is caused by hepatic

copper accumulation, we addressed the question whether the selenium status of these patients is impaired. We observed lower amounts of circulating SELENOP especially in patients with low serum CP concentrations which is an indicator for a more pronounced hepatic copper accumulation (Fig. 4e). So far, SELENOP concentrations have not been analyzed in serum samples of Wilson's patients. Only one study characterized the selenium status of Wilson's patients and found lower circulating selenium concentrations in male patients fitting to our SELENOP and selenium results being reduced in patients with lower circulating CP concentrations (Fig. 4e, f). This is also in line with the data obtained from the LPP rat model of Wilson's disease. In rats of the disease onset and disease groups, we observed a decline of both circulating selenium and SELENOP concentrations (Fig. 3d, e). Interestingly, a tendency for lower selenium concentrations was also observed in Wilson's patients with dominating neuropsychiatric symptoms[31], which could hint for reduced selenium transport to the brain via SELENOP. It has been well described that *SELENOP* knockout mice suffer from severe seizures and ataxia due to selenium deficiency

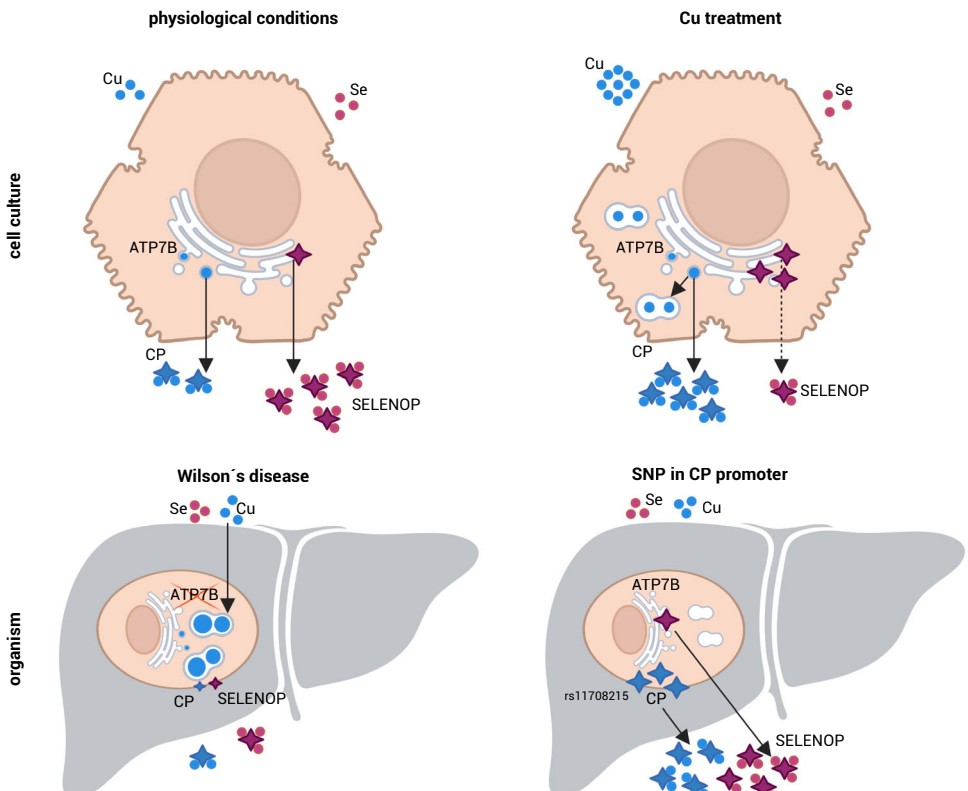

**Fig. 7 | Relationship between copper and selenium modulated by ceruloplasmin and SELENOP under different physiological and pathophysiological conditions.** Effects are depicted in a scheme as described in the text. The figure was created with BioRender: https://app.biorender.com/.

in the brain[5,6]. Based on this, it can be speculated that reduced selenium levels in the brain contribute to the development of neurological symptoms in Wilson's patients. In addition to Wilson's disease, cholestatic diseases such as primary biliary cholangitis, and primary sclerosing cholangitis (PSC) are accompanied by hepatic copper accumulation[32]. For example, in PSC hepatic selenium concentrations increase while serum selenium levels decrease[33], fitting to what we observed in Wilson's patients.

Under conditions of Wilson's disease, ATP7B-deficient hepatocytes lack the most effective defence mechanism against copper which is efficient copper export and therefore they have to rely on alternative mechanisms to survive copper overload that includes upregulation of copper binding proteins such as metallothionein or glutathione, allowing the cells to buffer increasing copper levels[34]. Also in our cell culture model, copper treatment resulted in transcriptional upregulation of metallothionein which was, however, less pronounced as compared to zinc-treated cells (Supplementary Fig. 2e). A further increase of copper exceeding the buffer capacity results in oxidative stress. SELENOP has been described to have antioxidant properties[8] and, thus, SELENOP accumulation could be an attempt to combat Cu-induced oxidative stress which would explain the necessity to stop its excretion under conditions of copper overload. We neither observed a difference in copper-induced cytotoxicity nor in total cellular copper concentrations upon *SELENOP* knockdown (Fig. 1l) indicating that SELENOP is not essential for limiting copper-induced oxidative stress in HepG2 cells. In line with this, *SELENOP* knockout mice do not show any difference in overall hepatic copper levels in comparison to wild type mice[35]. In LPP rats, oxidative stress is only a characteristic of the liver of the 'diseased' group and, thus, does not correlate with the reduced SELENOP excretion which was already observed in rats of the 'disease onset' group (Fig. 3e). Also the chosen cell culture conditions with a copper supply of 100 μM for up to 72 h do not result in oxidative

stress but only in a very mild Nrf2 activation[16]. This Nrf2 activation appears to result in the observed transcriptional changes with a copper- but also a zinc-induced downregulation of *SELENOP* and an upregulation of *LRP8* mRNA (Fig. 1f, Supplementary Fig. 2d) which both have been described to be mediated by Nrf2[36,37]. Accordingly, the accumulation of both intracellular selenium and SELENOP was not observed in cells treated with other redox-active metals such as iron or zinc or by treating cells with different concentrations of $H_2O_2$ (Fig. 2e) pointing towards a copper-specific mechanism which goes beyond a mere oxidative stress-mediated effect. Copper specificity was further supported by the reversal of the copper-induced SELENOP accumulation by co-treating with the copper chelators BCS or TTM (Fig. 2d) in HepG2 cells and by methanobactin in LPP rats (Fig. 3f–h). Thus, the copper-mediated SELENOP accumulation does not appear to be mainly driven by redox-dependent processes.

Prior to excretion, SELENOP is processed in the ER/Golgi-network[12]. The SELENOP molecule contains heparin binding regions, two histidine-rich domains and several N- and O-glycosylation sites which are post-translationally modulated[11,38]. The post-translational modifications are supposed to protect the Sec residues within SELENOP by reducing their reactivity. However, they are not modulating the amount of SELENOP excreted from HepG2 cells[12]. In contrast, heparin binding to SELENOP increased its excretion while APOE binding to the heparin binding regions reduced SELENOP excretion from hepatocytes. Accordingly, circulating APOE bound to SELENOP is involved in enhancing the binding of SELENOP to LRP8 on the surface of target cells[25]. While membrane localization of LRP8 was unaffected by copper (Supplementary Fig. 1f), we observed lower extracellular and higher intracellular APOE concentrations in copper-treated cells (Fig. 5a–c) which indicates an intracellular APOE accumulation. Copper has been well described to modulate lipid metabolism particularly cholesterol biosynthesis[34]. Copper can bind to APOE[39] and depending

on the APOE genotype, expression of metallothionein and, thus, intracellular copper binding capacity is modulated[40]. In addition, disruption of copper delivery to mitochondria increases APOE expression and secretion[41]. An *APOE* knockdown in HepG2 cells enhanced SELENOP release into the medium (Fig. 5e) which has been also shown before in the serum of *APOE* knockout mice[25]. Thus, APOE accumulation and binding to SELENOP provides a mechanistic link how copper could interfere with SELENOP excretion.

The ER/Golgi-network is also crucial for maintaining copper homeostasis as trafficking between membrane compartments is key to balance copper uptake and excretion[42]. ATOX1 ferries copper to ATP7A and ATP7B in the TGN for the metalation of newly synthesized copper-containing enzymes. Interestingly, this is the compartment in which SELENOP accumulates in response to copper treatment (Fig. 6f–h). But under conditions of copper exposure ATP7A/B translocate to post-Golgi membrane vesicles to either enhance copper efflux or to transiently store excess copper[43]. These processes are specific for hepatocytes because only in these cells a biliary copper secretion is possible. We did not observe a copper-induced selenium accumulation in intestinal cells (HT-29, Supplementary Fig. 1h) but rather a downregulation which is also the case in neurons (differentiated LUHMES cells) and partially in astrocytes (CCF-STTG1 cells)[44]. This indicates that the liver-specific shuttling of copper appears to be important for mediating SELENOP accumulation. Hepatic copper accumulation in Wilson's patients has been suggested to impair protein synthesis and general protein secretion indicated e.g., by the reduced release of functional coagulation factors[45]. Herein, no clear picture emerged regarding secretion of coagulation factors by HepG2 cells as some were less secreted but most of them were unaffected by copper (Supplementary Data 1). Based on our secretome data, we can exclude a general reduction of released proteins because the major part of the secreted proteins was unaffected by copper (including e.g., α1-antitrypsin; Supplementary Fig. 3c) and only 6.4% of the whole secretome were downregulated by copper. But obviously, there is an enrichment of glycoproteins in the group of proteins less secreted upon copper treatment (53% of the downregulated proteins are glycoproteins while only 40% of the upregulated proteins belong to this group; categorized based on GlyCosmos, https://glycosmos.org/glycoproteins/), which includes SELENOP and APOE[11,46]. Based on the results from the PNGase and EndoH digestion of lysates (Fig. 6c, d), it can be concluded that the copper-sensitive full glycosylation of SELENOP takes place in the late Golgi (Fig. 6e). It has been suggested that SELENOP glycosylation might function to prevent proteolysis of SELENOP, thus, stabilizing intracellular levels[47]. We observed no copper-induced intracellular SELENOP accumulation in cells treated with the glycosylation inhibitor tunicamycin (Supplementary Fig. 4e), indicating that this glycosylation is important for the observed copper-induced effects. Several proteins of the conserved oligomeric Golgi (COG) complex have been shown to interact with ATP7A/B indicating that Golgi homeostasis is modulated in response to the copper status. The association of ATP7 paralogs with the COG complex remained after challenging cells with copper[48,49]. But so far, no mechanism has been identified that explains how copper modulates Golgi shuttling and cargo sorting within cells. Just recently, it was proposed that copper-induced effects on mitochondria could be the main driver for the observed differences in the cellular secretome[50]. Accordingly, it appears most likely that hepatic SELENOP accumulation is driven by copper-induced modulation of intra-Golgi transfer and/or glycosylation processes resulting in SELENOP accumulation in the Golgi.

From the Wilson's disease samples (rat and human), we can conclude that a low serum copper concentration is positively correlated with low circulating SELENOP levels (humans: Pearson's $r = 0.425$; $p$-value = 0.07) which is supposed to be caused by hepatic copper accumulation. Another proxy for circulating copper is CP which is mainly secreted from liver and represents 80–95% of circulating copper in healthy individuals. The reduction of circulating holo-CP concentrations is one of the early symptoms of Wilson's disease caused by the impaired transfer of copper via ATP7B to CP and concomitant hepatic copper accumulation[51]. In the LPP rat model, serum levels of CP oxidase activity and SELENOP were positively correlated (Pearson's $r = 0.522$; $p$-value = 0.011). Several genetic polymorphisms have been described in the *CP* gene. One of those, SNP rs11708215 (minor allele frequency for GG: 0.19) located in the *CP* gene promoter was strongly associated with increased levels of circulating CP ($p = 9 \times 10^{-10}$) as compared to the AA genotype. The results were replicated in an independent cohort in which the relationships remained significant also after adjusting for potential covariates including cholesterol levels in blood[22]. In line with this, the analysis of EPIC-Potsdam samples revealed that serum levels of SELENOP were correlated with SNP rs11708215 (Fig. 4g) which has been shown to positively correlate with serum copper[23] and CP concentrations[22]. The effects of the genetic polymorphisms of *CP* on hepatic copper needs to be studied further but points towards a genetic component modulating the relationship between hepatic copper and SELENOP release (Fig. 7). Additional conditions resulting in the upregulation of CP expression and downregulation of hepatic copper concentrations are non-alcoholic fatty liver disease (NAFLD) and overnutrition[52]. In contrast to the described steady-state conditions, the secretome analyses herein identified a negative correlation of extracellular CP and SELENOP because of increased CP and decreased SELENOP release (Fig. 5a). Interestingly, additional negative acute phase proteins including transthyretin (FC of 0.72), thyroxine-binding protein (FC of 0.76), and alpha-fetoprotein (FC of 0.65) reacted in the same manner as SELENOP. Also, HepG2 cells treated with IL-6 start to enhance CP release while dampening extracellular SELENOP levels[10]. Thus, dynamics of adaptation processes regulating both copper and selenium homeostasis appear to be connected and modulated depending on the health status but also the genetic background of individuals.

## Methods

All research in this study complies with all relevant ethical regulations. Animal procedures were approved in case of mice by the Ministry of Environment, Health and Consumer Protection of the federal state of Brandenburg, Germany and conducted following national guidelines and institutional guidelines of the German Institute of Human Nutrition Potsdam-Rehbruecke, Germany. Animal experiments with LPP rats were approved by the government authorities of the Regierung von Oberbayern, Munich, Germany, and all animals were treated according to the guidelines for the care and use of laboratory animals of the Helmholtz Center Munich. Studies on Wilson's patients were approved by the Medical Ethics Committee of the University Hospital Heidelberg, Germany (Reference no. S-565/2011; DRKS00031527) and carried out in accordance with the declaration of Helsinki.

### Cell culture

Cell culture experiments with hepatocarcinoma-derived HepG2 from a male donor (ACC 180 German Collection of Microorganisms and Cell Cultures (DSMZ)) and adenocarcinoma-derived HT-29 cells from a female donor (ACC 299 DSMZ) were performed as previously described[16] with minor modifications. In addition to previously described experiments Cu(His)₃ (CuSO₄ (Merck, Darmstadt, Germany; 451657) and L-histidine (Merck; H6034) 1:3; pH 7.4) was used. The two different inhibitors of Golgi-ER shuttling, brefeldin A (0.1 µg/ml; Biomol, Hamburg, Germany; Cay11861) or monensin (0.5 µM; Sigma-Aldrich/Merck; M5273) and the inhibitor for protein glycosylation tunicamycin (1 µg/ml; Abcam, Cambridge, UK; ab120296) were added 24 h prior to harvest of the cells. ZnSO₄ (Merck; 1088830500) and FeCl₂ (Merck; 1038610250) were used in combination with selenite (Na₂SeO₃; Thermo Fisher Scientific; 15670510) for up to 72 h. Hydrogen peroxide (Merck; 1072090250) was added to selenium-supplied

cells for 6 h in phenol red-free RPMI (Thermo Fisher Scientific, Waltham, USA; 11835030) to induce oxidative stress. For secretome analyses, HepG2 cells were cultured for 48 h in Opti-MEM (Thermo Fisher Scientific; 11058-021) supplemented with 50 nM selenite. After indicated incubation times, media were collected and cells were harvested. Medium samples and cell pellets were frozen in liquid nitrogen and stored at −20 °C until further procedure.

### siRNA-mediated knockdown in HepG2 cells

HepG2 cells were transfected with either SEPP1 silencer select pre-designed siRNA (Ambion, Thermo Fisher Scientific, s12698), APOE silencer select pre-designed siRNA (Ambion, s536402) or silencer select negative control #1 siRNA (Ambion) as described by the manufacturer. Briefly, the transfection reagent lipofectamine 2000 (Thermo Fisher Scientific; 11668019) and the siRNAs were diluted in Opti-MEM (Thermo Fisher Scientific; 11058-021). The mixture of prediluted siRNA and lipofectamine was incubated at room temperature for 20 min and was added to the cells seeded in RPMI medium (Thermo Fisher Scientific; 21875091) without antibiotics at a final concentration of 1.5 µg or 5 pmol for lipofectamine and siRNA, respectively. Transfection was carried out for 18 h, before treatment with copper and/or selenium took place as described for additional 72 h. For cytotoxicity assays, transfected cells were pre-treated with selenite for 24 h and co-treated with selenite and increasing copper concentrations for additional 24 h.

### Cytotoxicity assay

For the MTT assay, cells were seeded in 96-well plates. After up to 72 h of incubation with the trace elements, the medium was discarded and medium without FCS containing 0.5 mg/ml thiazolyl blue tetrazolium bromide (MTT; Merck; M2128) was added to the wells. After 1 h, the medium was discarded and followed by a 10 min shaking step with DMSO (Carl Roth, Karlsruhe, Germany; 7029) to dissolve the obtained formazan crystals. Absorption was measured at 550 nm with 690 nm as reference wavelength, using a microplate reader (Synergy H1, Biotek, Bad Friedrichshall, Germany).

### Primary murine hepatocytes

Six male C57BL/6JCtrl mice were housed in a 12:12 h light/dark cycle at 22 °C and constant humidity at 55%, fed a nutrient-rich standard chow (V1534-300, Ssniff, Soest, Germany) ad libitum and were sacrificed in deep isoflurane (Abbott, Wiesbaden, Germany) anesthesia by cervical dislocation at the age of 20 weeks. Primary, mouse hepatocytes were isolated as described earlier[53]. After isolation and seeding, cells were treated with 50 nM $Na_2SeO_3$ in combination with or without 10 µM $CuSO_4$ for 24 h before harvesting.

### LPP rat model

Rat liver homogenate and plasma samples were generated and provided as previously described[54,55]. Briefly, rats were housed in a 12:12 h light/dark cycle at 22 °C and constant humidity at 55% and maintained on an ad libitum Altromin 1314 diet (Altromin Spezialfutter) and tap water. Some rats of the diseased group received the copper chelator methanobactin for 8 days[20]. Rats were anesthetized with isoflurane (Abbott) and blood was withdrawn to sacrifice the animals. Tissue samples were mixed with 0.1% Triton-X 100 (Carl Roth; 3051.4) and 2 µg/mL protease inhibitor (Merck; 539134) and centrifuged at 14,000 × g (10 min, 4 °C). The supernatant was used for further analysis.

### Wilson's patients

Serum samples were obtained from 23 patients at the time point of diagnosis of Wilson's disease recruited between 2011 and 2019 at the University Hospital Heidelberg, Germany as part of the clinical trial 'Biochemical and genetic markers of liver diseases'. Informed consent was obtained by participants. Patient characteristics are shown in table S1. Sex of participants was determined based on self-report.

Patients were classified into two groups (normal or low) based on their circulating CP concentration (90 µg/L as cut-off)[21]. Patients were further subdivided into 'severe disease' based on their inflammatory parameters such as serum CRP and hepatic transaminases as marker for liver pathology.

### Western Blot and Dot Blot

To prepare protein lysates, frozen cell pellets or tissues samples were homogenized in Tris buffer (100 mM Tris (Applichem, Darmstadt, Germany; A1086), 300 mM KCl (Applichem; 131494), pH 7.6 with 0.1% (v/v) Triton X-100 (Carl Roth; 3051.4)) using a TissueLyser II (Qiagen, Hilden, Germany) at 2 × 30 s maximum speed or lysed in RIPA buffer (50 mM Tris (Applichem; A1086), 150 mM NaCl (Carl Roth; 3957), 2 mM EDTA (Carl Roth; 8043.2), 0.5% sodium deoxycholate (Merck; D6750), 0.1% SDS (VWR, Radnor, USA; A1112,0500), 1% NP-40 alternative (Merck; 492016)), pH 7.7). Cellular debris was removed by centrifugation (14,000 × g, 10 min, 4 °C). For separation of cytosol and membrane fractions, the ProteoExtract® Subcellular Proteome Extraction Kit (Merck/Millipore; 539790) was used according to the manufacturer's instructions. Protein concentration was determined by Bradford analysis. Protein lysates or medium samples were mixed with 5x Laemmli-buffer (208.5 mM Tris (pH 6.8) (Applichem; A1086), 10% SDS (Applichem; A1112), 50% glycerin (Carl Roth; 6962.1), 12.5% β-mercaptoethanol (Merck; M3148), 0.625% bromphenol blue (Carl Roth; A512.1)) and boiled for 5 min at 95 °C. SDS polyacrylamide gel electrophoresis was followed by immunoblotting of proteins to nitrocellulose membranes. After immunoblotting, membranes were gently shaken for 2 min in Ponceau S solution (0.2% (w/v) Ponceau S (Carl Roth; 5938.2) with 3% (w/v) trichloroacetic acid (Carl Roth; 8789.2)) and bands were recorded by ChemiDoc™ MP Imaging System (Bio-Rad, Hercules, California, USA). Subsequently, membranes were blocked in 5% (w/v) non-fat dry milk in Tris-buffered saline containing 0.1% (v/v) Tween 20 (Carl Roth; 9127.1) (T-TBS) for 1 h at room temperature. The membranes were incubated with the following primary antibodies overnight at 4 °C: mouse anti-SELENOP (human samples; 1:5000[56],), rabbit anti-SELENOP (mouse and rat samples; 1:500-1:1000; immunoGlobe, Himmelstadt, Germany; 0122-03), rabbit anti-α1-antitrysin (1:5000; Abcam; 207303), rabbit anti-LRP8 (1:1500 in 5% BSA; Abcam; ab108208), rabbit anti-GAPDH (1:1000; Cell Signaling Technology, Danvers, MA, USA; 2118), rabbit GOLGIN-97 (1:1000; Cell Signaling Technology; 13192), and rabbit anti-ApoE (1:250; Invitrogen; 701241). As secondary antibody, horseradish peroxidase-conjugated goat anti-rabbit IgG (1:50000; Cell Signaling Technology; 7074S) or goat anti-mouse (1:3000; Cell Signaling Technology; 7076S) were incubated for up to 1.5 h in 5% (w/v) non-fat dry milk in T-TBS at room temperature. Proteins were detected using SuperSignal™ West Dura (Thermo Fisher Scientific; 34076) and band intensities were quantified densitometrically by the ChemiDoc™ MP Imaging System using Image Lab software version 5.0. Protein expression was normalized to Ponceau staining. Pictures of uncropped Western blots can be found in the Source data files. Due to high antibody specificity, dot blot analysis was used for the detection of human SELENOP in cell culture media. Samples were boiled for 5 min at 95 °C and 100 µl were loaded on T-TBS-rinsed nitrocellulose membranes by vacuum using a dot blot apparatus (Carl Roth).

### Deglycosylation using PNGase F and EndoH

HepG2 cell lysates were used for deglycosylation experiments using PNGase F (New England Biolabs, Frankfurt, German; P0704S) and EndoH (New England Biolabs; P0702S). 20 µg of protein was diluted with 10 × glycoprotein denaturing buffer and denatured at 100 °C for 10 min followed by cooling and a 10 s centrifugation step. Afterwards, 10 × GlycoBuffer 2 and 10% NP-40 (the latter only in case of PNGase F) were added to the mixture prior to the addition of 500 U enzyme and incubation for 1 h at 37 °C. A preparation without PNGase F or EndoH was used as negative control for each sample.

## Mass spectrometry analysis

Secretome samples were dissolved in 5% SDS in water and digested using the filter-aided sample preparation method[57]. In short, proteins were reduced with 100 mM dithiothreitol (Thermo Fisher Scientific; 165680250) at 60 °C for 30 min, transferred to 30 kDa Microcon Centrifugal Filter Units (Merck; MRCF0R030), washed several times with 8 M urea and once with digestion buffer (25 mM triethylammonium bicarbonate (TEAB) (Merck; T7408), 0.5% sodium deoxycholate (SDC) (Merck; D6750)) prior to alkylation with 10 mM methyl methanethiosulfonate (Thermo Fisher Scientific; 23011) in digestion buffer for 30 min. Samples were digested with trypsin (Pierce MS grade Trypsin, Thermo Fisher Scientific;90057; estimated ratio 1:20) at 37 °C overnight and two additional rounds of trypsin digestions were performed and incubated for two hours each. Peptides were collected by centrifugation. Digested peptides were purified using High Protein and Peptide Recovery Detergent Removal Spin Column (Thermo Fisher Scientific; 88306) according to the manufacturer instructions. SDC was removed by acidification with 10% trifluoroacetic acid (Thermo Fisher Scientific; 434161000). Samples were desalted (Pierce peptide desalting spin columns, Thermo Fisher Scientific; 89852) and dried prior to reconstitution in 3% acetonitrile (Thermo Fisher Scientific; 047138.M1) and 0.2% formic acid (FA) (Thermo Fisher Scientific; 270480250).

Samples were analyzed three times on an Orbitrap Fusion™ Tribrid™ mass spectrometer interfaced with Easy-nLC1200 liquid chromatography system (LC-MS/MS; Thermo Fisher Scientific). Peptides were trapped on an Acclaim Pepmap 100 C18 trap column (100 μm × 2 cm, particle size 5 μm; Thermo Fisher Scientific; 164213) and separated on an in-house packed analytical column (75 μm × 35 cm, particle size 3 μm; Reprosil-Pur C18; Dr. Maisch, Ammerbuch, Germany; r13.b9.) using a gradient from 5% to 80% acetonitrile in 0.2% FA over 90 min. Precursor ion mass spectra were acquired at 120,000 resolution, $m/z$ 400–1400 and maximum injection time of 50 ms. MS2 analysis was performed in a data-dependent mode using CID spectra of the 20 most intense precursor ions in each MS scan, recorded in the ion trap at collision energy setting of 35. Precursors were isolated in the quadrupole with a 1.2 $m/z$ isolation window, charge states 2 to 7 were selected for fragmentation and the dynamic exclusion was set to 20 s and 10 ppm.

Proteomic datasets were processed by MaxQuant ver. 1.6.10.43[58]. Data were searched against reference proteome of Homo Sapiens (101,702 entries, June 28 2022, UP000005640) downloaded from Uniprot. MaxQuant-implemented database was used for the identification of contaminants. Protein identification was done using these MaxQuant parameters as follows: mass tolerance for the first search 20 ppm, for the second search from recalibrated spectra 4.5 ppm; maximum of 2 missed cleavages; maximal charge per peptide $z = 7$; minimal length of peptide 7, maximal mass of peptide 4600 Da; carbamidomethylation (C) as fixed and acetylation (protein N-term) and oxidation (M) as variable modifications with the maximum number of variable modifications per peptide set to 5. Trypsin with no cleavage restriction was set as a protease. Mass tolerance for fragments in MS/MS was 0.5 Da, taking the 8 most abundant peaks per 100 Da for search (with enabled possibility of co-fragmented peptide identification). FDR filtering on peptide spectrum match was 0.01 and only proteins with at least one identified unique peptide were considered further. Proteins were quantified using MaxLFQ function[59] with at least one peptide ratio required for pair-wise comparisons of protein abundance between samples. Proteins identified as contaminants were removed before any further interpretation of data. Statistical analyses were performed in Perseus (v1.3.0.4). t test-based statistics were applied on normalized and logarithmized protein ratios to extract the significantly regulated proteins. Functional annotation of proteins by Gene Ontology (GO) was done using DAVID Gene Ontology enrichment analysis[60].

The mass spectrometry proteomics data have been deposited to the ProteomeXchange Consortium (https://proteomecentral. proteomexchange.org/cgi/GetDataset?ID=PXD036300) via the PRIDE partner repository[61] with the dataset identifier PXD036300.

## ELISA for SELENOP and CP in human serum samples

Human CP concentrations were measured using sandwich ELISA as described[18]. Briefly, serum samples of Morbus Wilson patients were pre-diluted 1:200 and 50 μl were used for CP quantification. A commercially available human CP standard (Ceruloplasmin 187-51-10, Lee Biosolutions, Maryland Heights, MO, USA) served as calibrator. The detection of an HRP-labeled antibody was performed applying 100 μl of 3,3′,5,5′-tetramethylbenzidine (TMB, Surmodics IVD, USA). The spectrophotometric read out was recorded within 10 min at 450 nm using a NanoQuant Infinite 200 Pro microplate reader (Tecan Group AG, Männerdorf, Schweiz). Serum SELENOP concentrations were measured using a validated, commercial sandwich ELISA (selenOtest™, selenOmed GmbH, Berlin, Germany). Sample preparation and assay procedure was performed according to manufacturer's instructions.

## CP oxidase activity

CP oxidase activity was measured in plasma samples of LPP rats and in serum samples of Wilson's patients which were either diluted 1:2 (ATP7B$^{+/-}$, Wilson's patients) or 1:5 (ATP7B$^{-/-}$ rats) in 0.9% NaCl (Carl Roth; 3957). 5 μl diluted plasma samples were mixed with 75 μl acetate buffer (0.1 M sodium acetate, pH 5; Roth; 3856) and were incubated for 5 min at 30 °C. Afterwards, 20 μl of a 7.88 mM o-dianisidine dihydrochloride solution (preheated to 30 °C) (VWR; A17175) were added to each well followed by another shaking step at 30 °C. For each sample, a background value was generated by adding 150 μl 9 M $H_2SO_4$ (VWR; 1.00731.2500) after 10 min which was subtracted from values generated by stopping the reaction after 60 min. The absorbance of the formed purple-red chromophore was measured at 550 nm with 690 nm as reference wavelength using a microplate reader (Synergy H1, Biotek) using GenS software version 3.11.

## qPCR

The mRNA of HepG2 cells was isolated with the Dynabeads mRNA DIRECT kit (Thermo Fisher Scientific; 61012) according to the manufacturer's description. The mRNA was reversely transcribed using the sensifast™ cDNA synthesis kit (Bioline Meridian Bioscience, Cincinnati, Ohio, USA; BIO-65054). Real-time PCRs were performed with 1x PerfeCTa SYBR Green Supermix (QuantaBio/VWR/ 733-1247) using cDNA-specific primers (Eurofins Genomics, Ebersberg, Germany) at a concentration of 250 nM in a total volume of 10 μl. Primer sequences were SELENOP (NM_005410) fwd: 5′-GAAACTCCATCGCCTCATTACCAT-3′; rev: 5′-CTGCCTATGCTGACCCTTGTG-3′; LRP8 (NM_004631.4): fwd: 5′-CACAAGCACATCTACTGGACTGAC-3′; rev: 5′-AGACCAATACATGAACCCTCGCAG-3′; MT2A (NM_005953. 3): fwd: 5′-AGGGCTGCATCTGCAAAGGG-3′; rev: 5′-TAGCAAACGGTCACGGTCAGGG-3′; RPL13A (NM_012423.2): fwd: 5′-AGCCTACAAGAAAGTTTGCCTATCTG-3′; rev: 5′-TAGTGG ATCTTGGCTTTCTCTTTCCT-3′. The Mx3005P QPCR System (Agilent Technologies, Santa Clara, CA, USA) was used with the following heating steps: 3 min at 95 °C, 40 cycles of 15 s at 95 °C, 20 s at 60 °C, and 30 s at 72 °C with all samples and standards measured in triplicates. Standard curves from diluted PCR products were used for quantification. Sample values were normalized to the reference gene RPL13A.

## Measurement of trace element concentrations

Trace element concentrations were measured in rat plasma and tissue samples, human serum, and cell lysates using a total reflection X-ray fluorescence (TXRF) spectrometer (Bruker Nano GmbH, Berlin, Germany) and TEsprit software version 1.0. As internal standard for cells

and rat tissue, 1 mg/L yttrium (Merck Millipore, Burlington, MA, USA; 119809) and for serum or plasma 1 mg/L gallium (Alfa Aesar/ Thermo Fisher Scientific, Kandel, Germany; 88066) was used. 10 μL of each sample were placed on siliconized (except plasma) quartz glass carriers and dried at 40 °C. Samples were measured in duplicates for up to 1000 s.

## Measurement of free copper
The concentration of free copper was quantified by a fluorometric method using a copper-binding fluorescent peptide (FP4) based on Young et al.[62]. and coupled to carboxyfluorescein as fluorophore. Serum samples were pre-diluted (1:20) in assay buffer (50 mM HEPES, pH 7.5[63],) and stored at −80 °C. An aliquot of 20 μL was added to 80 μL assay buffer containing FP4 (final concentration 10 nM) at room temperature. Free serum copper concentrations were calculated based on the fractional saturation of the sensor[64], determining the maximal and minimal fluorescence signal of FP4 after addition of 2 mM EDTA or 2.2 mM CuSO$_4$, respectively, and using a dissociation constant for the FP4-copper-complex of 0.37 pM.

## Association between candidate SNPs and SELENOP in EPIC-Potsdam
The EPIC-Potsdam cohort study consists of 27,548 participants recruited between 1994 and 1998 from the general population in Potsdam and surroundings, Germany. We used a random sample of 2500 individuals from 26,444 participants who provided blood samples at baseline. Of these, participants with prevalent diabetes, myocardial infarction or stroke were excluded. Further exclusion criteria were missing genetic data and missing data on trace element measurements, leaving 2204 individuals for analyses. DNA extraction from buffy coat, genotyping, and quality control within EPIC-Potsdam has been described in detail elsewhere[23]. Two candidate SNPs from the *CP* gene region were selected from previous GWAS analysis on serum copper levels in EPIC-Potsdam (rs11708215) and GWAS meta-analysis (rs34951015)[23]. Serum SELENOP levels were natural log-transformed to normalize the right-skewed distributions and standardized. We assumed an additive genetic model, adjusted for age at recruitment and sex.

## Immunofluorescence
Cells were grown on glass coverslip with the indicated trace elements for 24 h, fixed in 4% methanol-free paraformaldehyde (Thermo Fisher Scientific; 043368.9 L) for 10 min at room temperature and washed three times with PBS, permeabilized with Triton X-100 0.1% for 15 min at room temperature, rewashed with PBS and blocked with BSA 2% diluted in PBS Tween 20 0.1% (PBST) for 30 min at room temperature. Cells were incubated overnight at 4 °C with primary antibodies diluted in BSA 1% in PBST. The next day, cells were washed three times with PBS, incubated with fluorophore-conjugated secondary antibodies in 1% BSA-PBST for 1 h at room temperature, washed with PBS and then stained with DAPI (Thermo Fisher Scientific; D3571) for 5 min at room temperature, and mounted with ProLong™ glass anti-fade medium (Thermo Fisher Scientific; P3698). The following antibodies were used: mouse anti-SELENOP (1:200[56];), rabbit anti-GOLGIN97 (1:200; Abcam; Ab8340), and secondary antibodies anti-mouse AlexaFluor 488® (1:500; Thermo Fisher Scientific; A11029) and anti-Rabbit AlexaFluor 647® (1:500; Thermo Fisher Scientific; A32795).

## Statistics
Data are given as mean ± SD. The numbers of independent biological replicates are indicated in the figures as dots. Statistical analyses were performed using GraphPad Prism version 8 (San Diego, CA, USA). Mean differences between two groups were analyzed using Student's two-tailed *t* test. Differences between three or more groups were analyzed using one-way, two-way or three-way ANOVA followed by post hoc analysis through Bonferroni's multiple comparison test depending on the type of analyzed samples. Linear correlations were performed using the Pearson test. Data distribution was considered normal if it passed the Shapiro-Wilk normality test. A p-value below 0.05 was considered statistically significant.

## Reporting summary
Further information on research design is available in the Nature Portfolio Reporting Summary linked to this article.

## Data availability
The mass spectrometry proteomics data have been deposited to the ProteomeXchange Consortium (http://proteomecentral. proteomexchange.org) via the PRIDE partner repository[61] with the dataset identifier PXD036300. As reference proteome of *Homo Sapiens* UP000005640 was used. The remaining data generated in this study are provided in the Supplementary Information. Source data are provided with this paper.

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

## Acknowledgements

The authors highly acknowledge the excellent technical support by Stefanie Deubel, Alrun Schumann, Doreen Ziegenhardt, and Laura Taudte. We thank the Proteomics Core Facility at Sahlgrenska Academy, University of Gothenburg, Sweden for the protein quantification analysis. The Proteomics Core Facility is grateful to the Inga-Britt and Arne Lundbergs Forskningsstiftlese for the donation of the Orbitrap Fusion Tribrid MS instrument. This work was supported by the German Research Foundation (DFG) [FOR 2558; H.H., T.G., M.B.S., T.S., L.S., A.P.K.], the Carl Zeiss Foundation (IMPULS; A.P.K.), by the Swedish Research Council (2018-02318 and 2021-03138; V.S.) and by the Swedish Society for Medical Research (S18-034; V.S.). We are very grateful to E. for supporting this project.

## Author contributions

Conceptualization: M.S., K.L., T.S.; V.I.S., L.S., A.P.K.; Investigation: M.S., C.E.M., A.L., J.H., S.J., C.W., M.M., A.H., E.A.E, A.A.H.P., N.G., S.A., I.A.; Resources: C.O., I.M., H.H., T.G., M.B.S., U.M., H.Z.; Visualization: M.S., C.E.M., V.I.S.; Writing - original draft preparation: M.S., A.P.K.; Writing - review and editing: C.E.M., K.L., T.G., M.B.S., T.S., H.Z., V.I.S. L.S.; Funding Acquisition: H.H., T.G., M.B.S., T.S., V.I.S., L.S., A.P.K.; All authors have read and agreed to the published version of the manuscript.

## Funding

## Competing interests

L.S. holds shares of selenOmed GmbH, a company involved in selenium status assessment. The remaining authors declare no competing interests.

## Additional information

[1]Department of Nutritional Physiology, Institute of Nutritional Sciences, Friedrich Schiller University Jena, Dornburger Str. 24, 07743 Jena, Germany. [2]TraceAge-DFG Research Unit on Interactions of Essential Trace Elements in Healthy and Diseased Elderly, Potsdam-Berlin-Jena-Wuppertal, Germany. [3]Institute for Experimental Endocrinology, Charité - University Medical School Berlin, Hessische Straße 3-4, 10115 Berlin, Germany. [4]Department of Molecular Toxicology, German Institute of Human Nutrition Potsdam-Rehbrücke, Arthur-Scheunert-Allee 114-116, 14558 Nuthetal, Germany. [5]Department of Molecular Epidemiology, German Institute of Human Nutrition Potsdam-Rehbrücke, Arthur-Scheunert-Allee 114-116, 14558 Nuthetal, Germany. [6]Department of Internal Medicine IV, University Hospital Heidelberg, Im Neuenheimer Feld 672, 69120 Heidelberg, Germany. [7]Institute of Clinical Sciences, Department of Surgery, Sahlgrenska Center for Cancer Research, University of Gothenburg, Blå stråket 5, 41345 Gothenburg, Sweden. [8]Wallenberg Centre for Molecular and Translational Medicine, University of Gothenburg, 41345 Gothenburg, Sweden. [9]Department of Food Chemistry and Toxicology, Technical University Berlin, Gustav-Meyer-Allee 25, 13355 Berlin, Germany. [10]Institute of Biomedicine, Department of Microbiology and Immunology, University of Gothenburg, 41345 Gothenburg, Sweden. [11]The Institute of Medical Microbiology and Hygiene, University Medical Centre Freiburg, Freiburg, Germany. [12]Institute of Nutritional Science, University of Potsdam, Arthur-Scheunert-Allee 114-116, 14558 Nuthetal, Germany. [13]German Federal Institute for Risk Assessment (BfR), Max-Dohrn-Str. 8-10, 10589 Berlin, Germany. [14]Institute of Toxicology and Environmental Hygiene, Technical University Munich, School of Medicine, Biedersteinerstraße 29, 80802 Munich, Germany. [15]Institute of Molecular Toxicology and Pharmacology, Helmholtz Center Munich, German Research Center for Environmental Health, Ingolstädter Landstraße 1, 85764 Neuherberg, Germany. [16]These authors contributed equally: Maria Schwarz, Caroline E. Meyer. ✉e-mail: anna.kipp@uni-jena.de

