## [Peer Review File · Nature Communications]

Excessive copper impairs intrahepatocyte trafficking and secretion of selenoprotein PREVIEWER COMMENTS

Reviewer #1 (Remarks to the Author):

The researchers have studied the impact of hepatic copper on the hepatic synthesis of the selenium transport protein, selenoprotein P (SELENOP). The studies were precipitated by previous observations on low plasma levels of selenium in inflammations, and also some observations on low selenium values in Wilson disease. The aim of the study should be more clearly stated (li 116-117). The researchers used cell and animal models in their studies including a liver-derived cancer cell line (HepG2) and also primary murine hepatocytes in culture. In these models copper treatment led to intracellular retention of selenium and decreased extracellular SELENOP levels.

Accumulation of copper in liver is typically seen in Wilson's disease and in animal models of Wilson (LPP rats and Bedlington terriers). Plasma SELENOP levels were relatively low in LPP rats, especially in advanced derangement of liver functions with low ceruloplasmin values and presumably lowering of some other liver-synthesized proteins. The ceruloplasmin promotor gene contains the SNP rs11708215, that has been shown to be associated with increased plasma ceruloplasmin. Interestingly, an analysis of a cohort study (EPIC-Potsdam) showed a positive correlation of SNP rs11708215 and serum SELENOP levels, indicating a connection between ceruloplasmin and SELENOP synthesis.

In addition, the observations in the present study indicated a disrupting effect of copper excess on intracellular SELENOP trafficking and secretion.

However, it remains uncertain if the observed copper-selenium interactions takes place at the transcription / synthetic steps or at intra- or intra-to-extracellular release steps. Copper in excess as is seen in Wilson disease and various cholestatic diseases (PBC and PSC) is also known to inhibit non-specifically the liver synthesis of several proteins, eg. albumin. This involves the possibility that the inhibited SELENOP synthesis represents a nonspecific toxic effect.

The manuscript is well written, and might be accepted with minor revision.

Here, I would give some suggestions for manuscript improvement:

1. In the Introduction the authors could include (with references) cholestatic diseases with hepatic copper accumulation (primary biliary cholangitis, and primary sclerosing cholangitis), and possible effects on selenium metabolism:

Reference, for instance:

a) Dastyh, M., Husová, L., Aiglová, K., Fejfar, T., & Dastyh Jr, M. (2021). Manganese and copper levels in patients with primary biliary cirrhosis and primary sclerosing cholangitis. *Scandinavian Journal of Clinical and Laboratory Investigation*, 81(2), 116-120.

2. The authors should in Discussion more clearly include the possibility of non-specific inhibition on protein synthesis resulting from copper toxicity, as is seen by lowering of plasma levels of albumin and coagulation factors in Wilson (The secretion of these proteins from their cell cultures is insufficiently quantified).

References on the lowering of these proteins, for instance:

a) Członkowska, A., Litwin, T., Dusek, P., Ferenci, P., Lutsenko, S., Medici, V., ... & Schilsky, M. L. (2018). Wilson disease. *Nature reviews Disease primers*, 4(1), 1-20.

b) Schaefer, M., Weber, L., Gotthardt, D., Seessle, J., Stremmel, W., Pfeiffenberger, J., & Weiss, K. H. (2015). Coagulation Parameters in Wilson Disease. *Journal of Gastrointestinal & Liver Diseases*, 24(2).

3. The main secretory/excretory pathway of copper is via biliary secretion. Interactions between selenium and copper have previously been observed in studies of bile and biliary diseases, which could be mentioned in Discussion. Reference for instance:

Aaseth, J., Thomassen, Y., Aadland, E., Fausa, O., & Schruppf, E. (1995). Hepatic retention of copper and selenium in primary sclerosing cholangitis. *Scandinavian journal of gastroenterology*, 30(12), 1200-1203.

4. Another question is if lowered plasma selenium observed in advanced Wilson disease contribute to neurodegenerative disease, which in part is discussed by the authors. The synthesis of SELENOP in

CNS might also be impaired in copper loading diseases. Relevant references:

- a) Świątkowska-Stodulska, R., Dejneka, W., Owczarzak, A., Drobińska-Jurowiecka, A., Kiszkiś, H., Wiśniewski, P., & Sworczak, K. (2009). Assessment of selected oxidative stress parameters in patients with Wilson's disease. *Archives of Medical Science*, 5(3), 465-470.
- b) Solovyev, N., Drobyshev, E., Bjørklund, G., Dubrovskii, Y., Lysiuk, R., & Rayman, M. P. (2018). Selenium, selenoprotein P, and Alzheimer's disease: is there a link?. *Free Radical Biology and Medicine*, 127, 124-133.
- c) Yang, X., Hill, K. E., Maguire, M. J., & Burk, R. F. (2000). Synthesis and secretion of selenoprotein P by cultured rat astrocytes. *Biochimica et Biophysica Acta (BBA)-General Subjects*, 1474(3), 390-396.

Reviewer #2 (Remarks to the Author):

Dear authors,

It was a pleasure to go through your manuscript COPPER IMPAIRS INTRAHEPATOCYTE TRAFFICKING AND SECRETION OF SELENOPROTEIN P.

Summarising in short:

With the help of cell cultures (HepG2 cells, HT-29 cells), experimental animals (mouse liver cells, rat liver homogenate), and Willson disease patients (blood samples) and using various techniques the investigation confirmed and defined interactions between selenium, particularly selenoprotein P (SELENOP), and experimental or disease-related excessive copper levels. The inverse association was observed not only between copper and SELENOP but also between copper and apolipoprotein E and copper and ceruloplasmin (CP). The effects/interactions were tested by different methods including the untargeted secretome approach and CP polymorphism.

The quality of data from all points of view (techniques, data analyses, presentation, and interpretation) is in general excellent and they provide good evidence for your claims. Undoubtedly, the results and conclusions are important to the field and for a better understanding of basic selenium metabolism, its interactions with copper, and some other proteins involved in the case of excessive copper amounts.

Below I'm adding a few comments

1. The title is misleading as not all levels of copper are affecting the selenium metabolism in a similar way. Interactions could be different particularly in the case of copper deficiency. I suppose that because your investigation is related to excessive copper levels this should be addressed in the title.
2. In the discussion, I miss a sentence or two about the metallothioneins, which are known to be involved in Cu metabolism, particularly in Cu excess or deficiency. It seems that unfortunately, they have not been assessed according to the data presented in Table S2 (Secretome proteomics data of Copper stimulated HEPG2 cells compared to control). This absence should be commented on.
3. There are a few older studies reporting correlations between selenium and copper after supplementation with one or the other. In short term, such excessive exposure can lead to mutual detoxification (coaccumulation) which in prolongation triggers the deficiency (side effects) of a non-supplemented element.

Reviewer #3 (Remarks to the Author):

This study reveals a regulatory role for copper (Cu) in selenium (Se) metabolism. The authors demonstrate that elevated Cu levels inhibit secretion of selenoprotein P (SELENOP) from hepatic cells *in vitro* and *in vivo*. While the mechanisms operating in the uptake of SELENOP were well characterized, how SELENOP excretion is regulated remains unclear. The manuscript provides new insights into understanding this process and uncovers an important link between Cu and Se metabolism. Considering that both Se and Cu represent vitally important nutrients, whose imbalance causes severe symptoms, this study might be important for the characterization of new mechanisms driving pathogenesis of Cu- and Se-related metabolic disorders. Therefore, the manuscript would be of great interest to the broad readership of Nature Communications. However, in my view, the manuscript still lacks some mechanistic details and controls and should be revised to address the following comments.

1) Where is SELENOP retained upon elevated Cu?

The authors convincingly show that exposure to Cu reduces SELENOP secretion and causes its intracellular accumulation. However, it is unclear in which compartment SELENOP is retained. According to the authors, BFA and monensin cause a similar accumulation of SELENOP in hepatic cells but these drugs block cargo proteins at two different levels of the secretory pathway: BFA within the ER and monensin within the Golgi. To better understand the intracellular localization of SELENOP, I would suggest using immunofluorescent approaches to evaluate the overlap of SELENOP with ER and Golgi markers. If the SELENOP antibody does not work for IF, a tagged version of the protein might be expressed in HepG2 cells for these experiments. Alternatively, the authors could use subcellular fractionation to find the compartment that contains SELENOP in Cu-treated cells. Finally, an Endo-H approach could be employed for this purpose. The authors mention that SELENOP contains N-glycosylation sites. Endo-H digestion might reveal whether SELENOP contains early Golgi N-linked sugar chains, which are sensitive to Endo-H, or late Golgi Endo-H resistant N-linked sugars.

2) Can the authors rule out that Cu-mediated accumulation of SELENOP occurs due to elevated uptake by LRP8? Does Cu accelerate LRP8 expression? This can be checked by either Western blot or qRT-PCR.

3) Impact of other metals (Zn, Fe) on SELENOP secretion.

It would be interesting to evaluate SELENOP RNA levels by qRT-PCR to check whether the mechanism by which Zn and Fe impact on SELENOP excretion is different from Cu and whether any of them are related to transcriptional control of SELENOP.

4) *In vivo* impact of Cu on hepatic SELENOP excretion.

While the part of results with LLP rats is very straightforward, experiments in mice are less clear. Why did the authors use Cu depletion instead of Cu overload? Then the authors say that in the mice "... no effect was observed on hepatic selenium concentrations or on circulating SELENOP levels (Fig. 3A, B)." I suppose that Fig. 3A should show hepatic levels of selenium, but this panel is labelled as "serum". Finally, it is unclear why Se levels in the cerebellum are shown in panel C. It would be more informative to have hepatic Se levels.

5) The secretome data are very interesting and in an unbiased way support the main findings of the study. These data show that the SELENOP binding partner APOE was also retained in Cu-treated cells. Further, the authors demonstrated that silencing of APOE reduces SELENOP retention in Cu-treated cells. Taken together, these findings suggest that reduced SELENOP secretion might be caused by binding to APOE that accumulates in the Cu-treated cells. This however poses several obvious questions:

- Why Cu inhibits APOE secretion?
- Does Cu stimulate the interaction between SELENOP and APOE?
- Are SELENOP and APOE retained by Cu within the same compartment?

It would be of interest if the authors could address these points.

6) The relevance of the main findings for Wilson disease (WD) has to be discussed in the manuscript. Do reduced serum levels of SELENOP contribute to WD pathogenesis? Recent studies mentioned by the authors indicate that SELENOP deletion in mice results in severe seizures and ataxia due to selenium deficiency in the brain. Could reduced Se serum levels in WD patients contribute to the development of neurological symptoms?

Minor points.

a) Ponceau is used in all blots as a normalization/input marker. Can the authors explain this choice? Apparently, Cu does not affect levels of α 1-antitrypsin (AAT) in the medium/serum and, therefore, AAT could be used as a marker for normalization, while for cell lysates α -tubulin or GAPDH could be employed.

b) SELENOP should be in capital letters throughout the manuscript. In some parts of the text, it is written as "Selenop".

Point-to-Point reply

Reviewer #1:

The researchers have studied the impact of hepatic copper on the hepatic synthesis of the selenium transport protein, selenoprotein P (SELENOP). The studies were precipitated by previous observations on low plasma levels of selenium in inflammations, and also some observations on low selenium values in Wilson disease. The aim of the study should be more clearly stated (li 116-117).

We thank the reviewer for this important comment. We named the aim of the study more clearly in line 119-122:

Accordingly, we aimed to study whether copper excess affects intracellular selenium levels, and SELENOP distribution and excretion of cultured hepatocytes and whether the findings are compatible with data from genetic models of ATP7B mutations in rats and Wilson's disease patients.

The researchers used cell and animal models in their studies including a liver-derived cancer cell line (HepG2) and also primary murine hepatocytes in culture. In these models copper treatment led to intracellular retention of selenium and decreased extracellular SELENOP levels. Accumulation of copper in liver is typically seen in Wilson's disease and in animal models of Wilson (LPP rats and Bedlington terriers). Plasma SELENOP levels were relatively low in LPP rats, especially in advanced derangement of liver functions with low ceruloplasmin values and presumably lowering of some other liver-synthesized proteins. The ceruloplasmin promotor gene contains the SNP rs11708215, that has been shown to be associated with increased plasma ceruloplasmin. Interestingly, an analysis of a cohort study (EPIC-Potsdam) showed a positive correlation of SNP rs11708215 and serum SELENOP levels, indicating a connection between ceruloplasmin and SELENOP synthesis.

We thank the reviewer for this clear summary of our results. To improve clarity of the results for future readers we added a graphical abstract showing these main results and relationships between CP and SELENOP depending on the setting analysed.

In addition, the observations in the present study indicated a disrupting effect of copper excess on intracellular SELENOP trafficking and secretion. However, it remains uncertain if the observed copper-selenium interactions take place at the transcription / synthetic steps or at intra- or intra-to-extracellular release steps. Copper in excess as is seen in Wilson disease and various cholestatic diseases (PBC and PSC) is also known to inhibit non-specifically the liver synthesis of several proteins, eg. albumin. This involves the possibility that the inhibited SELENOP synthesis represents a nonspecific toxic effect.

We put a lot of effort in better pinpointing the mechanism underlying the copper-induced effects on SELENOP accumulation and added the following information:

Copper reduced SELENOP mRNA expression (new Fig. 1F) but only very mildly with a fold change of 0.8. Accordingly, an enhanced transcription does not appear to drive intracellular SELENOP accumulation. In line with this, we could show that mainly the fully glycosylated form of SELENOP accumulates in response to copper treatment (Fig. 1E). This glycosylation step takes place in the late Golgi and in this compartment SELENOP accumulation was observed after copper treatment (Fig. 6). For more detailed information see below.

The reviewer is right that Wilson's disease, especially during onset, is a rather extreme situation of copper accumulation. However, our cell culture setting was carefully validated and we are pretty sure that we do not produce cytotoxic effects. This is also supported by the secretome data, which does not provide evidence for a general inhibition of protein synthesis

by copper treatment. More than twice as much proteins showed a higher secretion in response to copper and only 75 proteins were secreted less. This is now more extensively discussed (line 406-413).

Hepatic copper accumulation in Wilson's patients has been suggested to impair protein synthesis and general protein secretion indicated e.g., by the reduced release of functional coagulation factors [45]. Herein, no clear picture emerged regarding secretion of coagulation factors by HepG2 cells as some were less secreted but most of them were unaffected by copper (Tab. S2). Based on our secretome data, we can exclude a general reduction of released proteins because the major part of the secreted proteins was unaffected by copper (including e.g., α 1-antitrypsin; Fig. S3C) and only 6.4 % of the whole secretome were downregulated by copper.

The manuscript is well written, and might be accepted with minor revision. Here, I would give some suggestions for manuscript improvement:

1. In the Introduction the authors could include (with references) cholestatic diseases with hepatic copper accumulation (primary biliary cholangitis, and primary sclerosing cholangitis), and possible effects on selenium metabolism:

Reference, for instance:

a) Dastyh, M., Husová, L., Aiglová, K., Fejfar, T., & Dastyh Jr, M. (2021). Manganese and copper levels in patients with primary biliary cirrhosis and primary sclerosing cholangitis. *Scandinavian Journal of Clinical and Laboratory Investigation*, 81(2), 116-120.

We added this important information and the suggested reference to the manuscript, line 342-346:

In addition to Wilson's disease, cholestatic diseases such as primary biliary cholangitis, and primary sclerosing cholangitis (PSC) are accompanied by hepatic copper accumulation [32]. For example, in PSC hepatic selenium concentrations increase while serum selenium levels decrease [33], fitting to what we observed in Wilson's patients.

2. The authors should in Discussion more clearly include the possibility of non-specific inhibition on protein synthesis resulting from copper toxicity, as is seen by lowering of plasma levels of albumin and coagulation factors in Wilson (The secretion of these proteins from their cell cultures is insufficiently quantified).

As discussed above, at least in the cell culture setting, we did not observe cytotoxic effects of 100 μ M copper. Unfortunately, albumin was not detected by the secretome approach but other proteins such as α 1-antitrypsin with also high hepatic secretion rate were completely unaffected by copper (Fig. S3C). For coagulation factors, the regulation pattern by copper was very heterogeneous as detected by the secretome approach, but most of the classical coagulation factors were not modulated by copper:

UniProt Accession ID	Protein names	Gene names	-log10(p-val)	Fold change (Se+Cu/Se)
H7BZ18	Multiple coagulation factor deficiency protein 2	MCFD2	NaN	1.000
X6R3B1	Coagulation factor XI;Coagulation factor XIa heavy chain;Coagulation factor XIa light chain	F11	NaN	1.000
P00742	Coagulation factor X;Factor X light chain;Factor X heavy chain;Activated factor Xa heavy chain	F10	0.659	0.796
A0A0A0MRJ7	Coagulation factor V;Coagulation factor V heavy chain;Coagulation factor V light chain	F5	0.892	0.880
Q6PCB0	von Willebrand factor A domain-containing protein 1	VWA1	2.471	0.597
Q5GFL6-3	von Willebrand factor A domain-containing protein 2	VWA2	0.752	0.782
P02671-2	Fibrinogen alpha chain;Fibrinopeptide A;Fibrinogen alpha chain	FGA	2.129	1.847
C9JEU5	Fibrinogen gamma chain	FGG	5.852	0.555
P02671	Fibrinogen alpha chain;Fibrinopeptide A;Fibrinogen alpha chain	FGA	4.956	0.582
P02675	Fibrinogen beta chain;Fibrinopeptide B;Fibrinogen beta chain	FGB	4.202	0.639
P02751-1	Fibronectin;Anastellin;Ugl-Y1;Ugl-Y2;Ugl-Y3	FN1	1.782	1.065
P00734	Prothrombin;Activation peptide fragment 1;Activation peptide fragment 2;Thrombin light chain;Thrombin heavy chain	F2	4.468	0.654
P01008	Antithrombin-III	SERPINC1	1.437	0.913
G5E9F8	Vitamin K-dependent protein S	PROS1	0.585	0.933
E7END6	Vitamin K-dependent protein C;Vitamin K-dependent protein C light chain;Vitamin K-dependent protein C heavy chain	PROC	0.977	1.174

Nevertheless, we mentioned this important aspect in the discussion. See line 406-410:

Hepatic copper accumulation in Wilson's patients has been suggested to impair protein synthesis and general protein secretion indicated e.g. by the reduced release of functional coagulation factors [45]. Herein, no clear picture emerged regarding secretion of coagulation factors by HepG2 cells as some were less secreted but most of them were unaffected by copper (Tab. S2).

References on the lowering of these proteins, for instance:

- a) Cżłonkowska, A., Litwin, T., Dusek, P., Ferenci, P., Lutsenko, S., Medici, V., ... & Schilsky, M. L. (2018). Wilson disease. Nature reviews Disease primers, 4(1), 1-20.
- b) Schaefer, M., Weber, L., Gotthardt, D., Seessle, J., Stremmel, W., Pfeiffenberger, J., & Weiss, K. H. (2015). Coagulation Parameters in Wilson Disease. Journal of Gastrointestinal & Liver Diseases, 24(2).

We included the connection of copper and coagulation factors as stated above.

3. The main secretory/excretory pathway of copper is via biliary secretion. Interactions between selenium and copper have previously been observed in studies of bile and biliary diseases, which could be mentioned in Discussion. Reference for instance:

Aaseth, J., Thomassen, Y., Aadland, E., Fausa, O., & Schrupf, E. (1995). Hepatic retention of copper and selenium in primary sclerosing cholangitis. Scandinavian journal of gastroenterology, 30(12), 1200-1203.

We included this reference as [33] as described above.

4. Another question is if lowered plasma selenium observed in advanced Wilson disease contribute to neurodegenerative disease, which in part is discussed by the authors. The synthesis of SELENOP in CNS might also be impaired in copper loading diseases. Relevant references:

- a) Świątkowska-Stodulska, R., Dejneka, W., Owczarzak, A., Drobińska-Jurowiecka, A., Kiszki, H., Wiśniewski, P., & Sworczak, K. (2009). Assessment of selected oxidative stress parameters in patients with Wilson's disease. Archives of Medical Science, 5(3), 465-470.
- b) Solovyev, N., Drobyshev, E., Bjørklund, G., Dubrovskii, Y., Lysiuk, R., & Rayman, M. P. (2018). Selenium, selenoprotein P, and Alzheimer's disease: is there a link?. Free Radical Biology and Medicine, 127, 124-133.
- c) Yang, X., Hill, K. E., Maguire, M. J., & Burk, R. F. (2000). Synthesis and secretion of selenoprotein P by cultured rat astrocytes. Biochimica et Biophysica Acta (BBA)-General Subjects, 1474(3), 390-396.

We thank the reviewer for this important comment. We have data indicating that selenium accumulation in response to copper does not happen to the same extent in astrocytes and

neurons as observed here in liver-derived cells. These data were just recently published (Raschke et al., 2023, JTEMB). We included this information in the discussion, lines 402-405:

We did not observe a copper-induced selenium accumulation in intestinal cells (HT-29, Fig. S1H) but rather a downregulation which is also the case in neurons (differentiated LUHMES cells) and partially in astrocytes (CCF-STTG1 cells) [44].

Reviewer #2 (Remarks to the Author):

Dear authors,

It was a pleasure to go through your manuscript COPPER IMPAIRS INTRAHEPATOCYTE TRAFFICKING AND SECRETION OF SELENOPROTEIN P.

Thank you very much for this positive feedback.

Summarising in short:

With the help of cell cultures (HepG2 cells, HT-29 cells), experimental animals (mouse liver cells, rat liver homogenate), and Willson disease patients (blood samples) and using various techniques the investigation confirmed and defined interactions between selenium, particularly selenoprotein P (SELENOP), and experimental or disease-related excessive copper levels. The inverse association was observed not only between copper and SELENOP but also between copper and apolipoprotein E and copper and ceruloplasmin (CP). The effects/interactions were tested by different methods including the untargeted secretome approach and CP polymorphism.

The quality of data from all points of view (techniques, data analyses, presentation, and interpretation) is in general excellent and they provide good evidence for your claims. Undoubtedly, the results and conclusions are important to the field and for a better understanding of basic selenium metabolism, its interactions with copper, and some other proteins involved in the case of excessive copper amounts.

Below I'm adding a few comments

1. The title is misleading as not all levels of copper are affecting the selenium metabolism in a similar way. Interactions could be different particularly in the case of copper deficiency. I suppose that because your investigation is related to excessive copper levels this should be addressed in the title.

This is a very valid point which we clarified both in the title but also in the abstract, see lines 2 and 63:

Excessive copper impairs intrahepatocyte trafficking and secretion of selenoprotein P

2. In the discussion, I miss a sentence or two about the metallothioneins, which are known to be involved in Cu metabolism, particularly in Cu excess or deficiency. It seems that unfortunately, they have not been assessed according to the data presented in Table S2 (Secretome proteomics data of Copper stimulated HEPG2 cells compared to control). This absence should be commented on.

The reviewer is right. MTs are very important to bind excess intracellular copper as mentioned in the discussion (lines 343-349). MTs are primarily intracellular proteins and are not expected to show up in the secretome. We included qPCR data confirming the expected

upregulation of MT in response to copper or zinc treatment, indicating that the cellular adaptation mechanisms are obviously working (see Fig. S2E).

3. There are a few older studies reporting correlations between selenium and copper after supplementation with one or the other. In short term, such excessive exposure can lead to mutual detoxification (coaccumulation) which in prolongation triggers the deficiency (side effects) of a non-supplemented element.

We thank the reviewer for this valuable comment. However, in our setting such a high supplementation was not the focus of our research. We think that based on our cell culture results we can exclude pure unspecific side effects. In the liver (and e.g. the kidney of LPP rats) we do not observe a negative correlation of both elements but rather a positive one.

Reviewer #3 (Remarks to the Author):

This study reveals a regulatory role for copper (Cu) in selenium (Se) metabolism. The authors demonstrate that elevated Cu levels inhibit secretion of selenoprotein P (SELENOP) from hepatic cells in vitro and in vivo. While the mechanisms operating in the uptake of SELENOP were well characterized, how SELENOP excretion is regulated remains unclear. The manuscript provides new insights into understanding this process and uncovers an important link between Cu and Se metabolism. Considering that both Se and Cu represent vitally important nutrients, whose imbalance causes severe symptoms, this study might be important for the characterization of new mechanisms driving pathogenesis of Cu- and Se-related metabolic disorders. Therefore, the manuscript would be of great interest to the broad readership of Nature Communications. However, in my view, the manuscript still lacks some mechanistic details and controls and should be revised to address the following comments.

1) Where is SELENOP retained upon elevated Cu?

The authors convincingly show that exposure to Cu reduces SELENOP secretion and causes its intracellular accumulation. However, it is unclear in which compartment SELENOP is retained. According to the authors, BFA and monensin cause a similar accumulation of SELENOP in hepatic cells but these drugs block cargo proteins at two different levels of the secretory pathway: BFA within the ER and monensin within the Golgi. To better understand the intracellular localization of SELENOP, I would suggest using immunofluorescent approaches to evaluate the overlap of SELENOP with ER and Golgi markers.

If the SELENOP antibody does not work for IF, a tagged version of the protein might be expressed in HepG2 cells for these experiments. Alternatively, the authors could use subcellular fractionation to find the compartment that contains SELENOP in Cu-treated cells.

Finally, an Endo-H approach could be employed for this purpose. The authors mention that SELENOP contains N-glycosylation sites. Endo-H digestion might reveal whether SELENOP contains early Golgi N-linked sugar chains, which are sensitive to Endo-H, or late Golgi Endo-H resistant N-linked sugars.

The reviewer is totally right that the identification of the compartment where SELENOP accumulates upon copper treatment is crucial. The suggestions for methods to address this point were very helpful. First, we started to more clearly differentiate the results obtained by monensin and BFA treatment (new Fig. 6A, B and Fig. S4A-D) as described in the results section, lines 278-291:

As secretory proteins are processed in the Golgi, we used the inhibitors brefeldin A and monensin to interfere with intracellular protein shuttling. Brefeldin A blocks the transport of secretory proteins from the ER to the Golgi complex, and thus completely abolishes Golgi-resident glycosylation processes [26]. Accordingly, no distinct glycosylation of intracellular SELENOP was detectable in brefeldin A-treated cells (Fig. 6A). There was no copper-dependent effect on intracellular SELENOP or selenium levels after brefeldin A treatment (Fig. S4A, B). Next, monensin was used to inhibit protein transport from the medial to the trans Golgi complex [27]. Monensin-treated cells showed strong intracellular SELENOP accumulation which, however, was restricted to the two lower bands of SELENOP (Fig. 6B). The fully glycosylated form of SELENOP with a size of 65 kDa which was most sensitive to copper treatment (Fig. 1D, E) was not detectable after monensin treatment indicating that this glycosylation step takes place in the trans or late Golgi. In line with this, copper treatment did not increase intracellular SELENOP but even decreased SELENOP and selenium levels in monensin-treated cells (Fig. S4C, D).

Next, we added the suggested Endo-H digestion to the PNGase treatment (new Fig. 6C, D). EndoH digestion clearly showed that the copper-sensitive glycosylation is EndoH resistant as described in the results section, lines 291-301:

To further specify the compartment where the copper-sensitive glycosylation of SELENOP takes place, deglycosylation experiments using PNGase F and EndoH were performed [28]. PNGase F digestion cleaves off all N-glycans and accordingly results in the deglycosylation of all three SELENOP forms (Fig. 6C). A comparable lack of glycosylation was observed when treating cells with tunicamycin which inhibits N-linked glycosylation [29] and accordingly also substantially reduced SELENOP glycosylation (Fig. S4E). In tunicamycin-treated cells, copper reduced intracellular SELENOP levels (Fig. S4E) as observed in monensin-treated cells. In contrast, EndoH more specifically cleaves off early Golgi N-linked sugar chains, while late Golgi N-linked sugars are EndoH resistant. The copper-sensitive glycosylation of SELENOP turned out to be EndoH resistant (Fig. 6D) and accordingly appears to be established in the late Golgi (Fig. 6E).

Cellular fractionation experiments revealed that SELENOP accumulated in the membrane/organelle fraction enriched for the Golgi marker Golgin97 (new Fig. 6F). Together with co-localization experiments of SELENOP and Golgin97 (new Fig. 6G) also immune fluorescence experiments showed that SELENOP accumulates in the Golgi in response to copper treatment as described in the results section, lines 301-308:

Cellular fractionation experiments revealed that the Cu-induced SELENOP accumulation was undetectable in the cytosol but was enriched in the membrane/organelle fraction which also contained GOLGIN97, an established marker for the trans-Golgi network (TGN) (Fig. 6F). This accumulation was detectable already after 1 h of copper treatment and increased over time (Fig. S4F). In parallel with SELENOP, glycosylated APOE increased in the membrane/organelle fraction (Fig. S4G). Immune fluorescence experiments also identified a copper-induced increase of SELENOP in the perinuclear region showing a co-localization with GOLGIN97 (Fig. 6G, H).

2) Can the authors rule out that Cu-mediated accumulation of SELENOP occurs due to elevated uptake by LRP8? Does Cu accelerate LRP8 expression? This can be checked by either Western blot or qRT-PCR.

This is a very valid concern that we tried to address by first analysing LRP8 mRNA expression in response to copper (Fig. 1F). LRP8 mRNA expression is upregulated by a factor of 1.3 which is most probably mediated by Nrf2. Copper-mediated Nrf2 activation has been previously shown by us (Schwarz et al., 2020, Redox Biol.) and it is supposed that LRP8 belongs to the group of Nrf2 target genes. However, for transport proteins mRNA expression might be less important because localization at the membrane is most conclusive with respect to its ability to act as a functional transport protein. Therefore, we analysed LRP8 levels by Western Blot in the membrane fraction after copper treatment. There was no difference in membrane localization of LRP8 in relation to copper treatment (Fig. S1F). Based on this, we exclude that copper substantially modulates re-uptake of SELENOP from the culture medium.

3) Impact of other metals (Zn, Fe) on SELENOP secretion. It would be interesting to evaluate SELENOP RNA levels by qRT-PCR to check whether the mechanism by which Zn and Fe impact on SELENOP excretion is different from Cu and whether any of them are related to transcriptional control of SELENOP.

We included qPCR results for SELENOP and LRP8 for all three trace elements (Cu, Zn, Fe) as new Fig. S2D. As described above, we observed a mild downregulation of SELENOP and upregulation of LRP8 in response to copper treatment. Both effects were even more pronounced after incubation with zinc, while there was no effect on both genes after iron treatment. This observation further supports the idea that Nrf2 is the mediating transcription factor, because zinc is a well described Nrf2 activator and downregulation of SELENOP and upregulation of LRP8 have been described in response to Nrf2 activation. As there was no intracellular SELENOP accumulation in response to zinc but in response to iron, the transcriptional regulation of SELENOP and LRP8 (comparable for Cu and Zn) does not appear to be of major relevance for intracellular SELENOP accumulation.

4) In vivo impact of Cu on hepatic SELENOP excretion. While the part of results with LLP rats is very straightforward, experiments in mice are less clear. Why did the authors use Cu depletion instead of Cu overload? Then the authors say that in the mice "... no effect was observed on hepatic selenium concentrations or on circulating SELENOP levels (Fig. 3A, B)." I suppose that Fig. 3A should show hepatic levels of selenium, but this panel is labelled as "serum". Finally, it is unclear why Se levels in the cerebellum are shown in panel C. It would be more informative to have hepatic Se levels.

We agree with the reviewer that the mouse data is not so conclusive as hepatic copper concentrations were not modulated by the dietary intervention. In the meantime, we tried to establish a dietary copper overload in mice but were unsuccessful as dietary copper concentrations up to 30-fold of the recommendation were well tolerated by the mice and did not result in higher hepatic copper concentrations. Based on this, we decided to exclude the mouse data from the manuscript and focussed more on the LPP rat model. The data from LPP rats were further strengthened by including data on methanobactin-treated rats, a copper chelator which normalized circulating copper levels. Accordingly, serum selenium and SELENOP levels increased (Fig. 3F-H).

5) The secretome data are very interesting and in an unbiased way support the main findings of the study. These data show that the SELENOP binding partner APOE was also retained in Cu-treated cells. Further, the authors demonstrated that silencing of APOE reduces SELENOP retention in Cu-treated cells. Taken together, these findings suggest that reduced SELENOP secretion might be caused by binding to APOE that accumulates in the Cu-treated cells. This however poses several obvious questions:

- Why Cu inhibits APOE secretion?
- Does Cu stimulate the interaction between SELENOP and APOE?
- Are SELENOP and APOE retained by Cu within the same compartment?

It would be of interest if the authors could address these points.

We thank the reviewer for the positive feedback on the secretome data. It has been previously described that SELENOP secretion is negatively regulated by the interaction with APOE, which interacts with the H-rich domains of SELENOP (Jin et al., 2019). APOE follows the classical secretory pathway, in which synthesis in the ER is followed by movement through the Golgi and trans-Golgi network, during which APOE is glycosylated and sialylated. It is unclear so far, how copper inhibits APOE secretion and if copper stimulates the interaction between SELENOP and APOE. But we can clearly show that glycosylation of APOE is modulated in response to copper in a comparable manner as for SELENOP. But the effect on APOE glycosylation was clearly restricted to the membrane/organelle fraction and was neither observed in cytosol nor in whole cell lysates. Accordingly, both, SELENOP and APOE, accumulate in the membrane/organelle fraction enriched for the Golgi compartment in parallel over time (Fig. S4F, G). Overall, we observed an enrichment of glycoproteins in the group of proteins downregulated by copper in comparison to those upregulated by copper (see lines 414-424). This implies that copper might modulate glycosylation processes in the late Golgi.

6) The relevance of the main findings for Wilson disease (WD) has to be discussed in the manuscript. Do reduced serum levels of SELENOP contribute to WD pathogenesis? Recent studies mentioned by the authors indicate that SELENOP deletion in mice results in severe seizures and ataxia due to selenium deficiency in the brain. Could reduced Se serum levels in WD patients contribute to the development of neurological symptoms?

This aspect has been described more clearly in the discussion, lines 340-343:

It has been well described that SELENOP knockout mice suffer from severe seizures and ataxia due to selenium deficiency in the brain [5,6]. Based on this, it can be speculated that reduced selenium levels in the brain contribute to the development of neurological symptoms in Wilson's patients.

Minor points.

a) Ponceau is used in all blots as a normalization/input marker. Can the authors explain this choice? Apparently, Cu does not affect levels of α 1-antitrypsin (AAT) in the medium/serum and, therefore, AAT could be used as a marker for normalization, while for cell lysates α -tubulin or GAPDH could be employed.

The reviewer is right. Nevertheless, we decided to use Ponceau staining as this produced the most stable results and could be consistently used for normalization of all blots independent of the sample analysed (medium, serum, cell lysate...).

b) SELENOP should be in capital letters throughout the manuscript. In some parts of the text, it is written as "Selenop".

We adjusted the writing of SELENOP throughout the whole manuscript.

REVIEWERS' COMMENTS

Reviewer #1 (Remarks to the Author):

The interference of copper with the synthesis/ export of selenoprotein P (SELENOP), which is disclosed and mapped, is of great interest, and will be of significance for further studies in this area. The presented results support the conclusions. The methodology is sound and well described. My previous concerns have been adequately addressed in the revision.

Reviewer #3 (Remarks to the Author):

The authors seriously addressed all my comments and concerns during revision of the manuscript, which was significantly improved over the original version. I suggest to accept revised version for publication.